# Structure-guided loop grafting improves expression and stability of influenza neuraminidase for vaccine development

Pramila Rijal[1,2]*, Leiyan Wei[1,2], Guido C Paesen[3], David I Stuart[1,3], Mark Haworth[4], Kuan-Ying A Huang[5,6]*, Thomas A Bowden[3], Alain RM Townsend[1,2]*

[1]Chinese Academy of Medical Science Oxford Institute, Nuffield Department of Medicine, University of Oxford, Oxford, United Kingdom; [2]MRC Human Immunology Unit, MRC Weatherall Institute of Molecular Medicine, John Radcliffe Hospital, University of Oxford, Oxford, United Kingdom; [3]Division of Structural Biology, Centre for Human Genetics, University of Oxford, Oxford, United Kingdom; [4]Department of Pharmacology, University of Cambridge, Cambridge, United Kingdom; [5]Graduate Institute of Immunology and Department of Paediatrics, National Taiwan University Hospital, College of Medicine, National Taiwan University, Taipei, Taiwan; [6]Genomics Research Center, Academia Sinica, Taipei, Taiwan

**\*For correspondence:**
pramila.rijal@rdm.ox.ac.uk (PR);
arthurhuang1726@ntu.edu.tw
(K-YAH);
alain.townsend@imm.ox.ac.uk
(ARMT)

## eLife Assessment

The authors developed a methodology to graft antigenic surface loops on influenza virus neuraminidases. The hybrid proteins retained the structure of the neuraminidase scaffold and the antigenicity of the grafted loops. This **fundamental** work should help in developing novel neuraminidase constructs for use in influenza virus vaccines. The paper presents **compelling** evidence supporting the conclusions arrived at by the authors.

**Abstract** Influenza virus neuraminidase (NA) is a crucial target for protective antibodies, yet the development of recombinant NA protein as a vaccine has been held back by instability and variable expression. We have taken a pragmatic approach to improving expression and stability of NA by grafting antigenic surface loops from low-expressing NA proteins onto the scaffold of high-expressing counterparts. The resulting hybrid proteins retained the antigenic properties of the loop donor while benefiting from the high-yield expression, stability, and tetrameric structure of the loop recipient. These hybrid proteins were recognised by a broad set of human monoclonal antibodies elicited by influenza infection or vaccination, with X-ray structures validating the accurate structural conformation of the grafted loops and the enzymatic cavity. Immunisation of mice with NA hybrids induced inhibitory antibodies to the loop donor and protected against lethal influenza challenge. This pragmatic technique offers a robust solution for improving the expression and stability of influenza NA proteins for vaccine development.

## Introduction

Influenza virus neuraminidase (NA) is an important target for protective antibodies and therapeutic drugs against influenza viruses. Currently, influenza subunit vaccines focus on haemagglutinin and contain minimal amounts of NA, despite emerging evidence from multiple studies indicating NA's importance as an independent protective antigen (*Kilbourne et al., 1995*; reviewed in *Air, 2012*;

*Eichelberger et al., 2018*; *Giurgea et al., 2020*; *Rajendran et al., 2021*; *Zhang and Ross, 2024*). However, recombinant NA protein expression is often bedevilled by low yield and poor stability, with substantial variation in NA protein yields across different virus strains (*Supplementary file 1*; *Ellis et al., 2022*; *Prevato et al., 2015*; *Ecker et al., 2020*; *Martinet et al., 1997*; *van der Woude et al., 2020*; *Subathra et al., 2014*; *Schmidt et al., 2011*; *Nivitchanyong et al., 2011*; *Margine et al., 2013*; *Liu et al., 2015*). With the ExpiCHO transient expression system (Thermo Fisher), we have found that yields of the N1 subtype NA varied from undetectable (H1N1/2019) to a maximum of 380 mg/l of culture (H5N1/2021).

We have applied structural and epitope information to devise a solution to improve the expression and stability of our recombinant NAs (*Figure 1*, *Figure 1—figure supplement 1*). NA is a mushroom-shaped type II homotetrameric protein (*Varghese et al., 1983*; reviewed in *Air, 2012*). The polypeptide chain of the NA monomeric head folds into six, topologically identical, four-stranded, antiparallel β-sheets which are arranged like the blades of a propeller (*Varghese et al., 1983*). Each of the six β-sheet blades, numbered B1–B6, is composed of four antiparallel β-strands in a W-shape numbered S1–S4. The β-strands in the 'W' are connected at the top and bottom by loops (*Figure 1a*). The loop between the fourth strand of the preceding β-sheet and the first strand of the following sheet (L01), and the loop between the second and third strands (L23) form most of the top surface and the active site of the NA and were delineated in the first crystal structure of the N2 NA (*Varghese et al., 1983*; *Colman et al., 1983*) and in subsequent structures (reviewed in *Air, 2012*). As there are six β-sheets in each NA monomer, there are six L01 and six L23 loops that together form most of the top surface of each identical NA monomer with a final contribution from the C-terminal domain (CTD) of ~17 amino acids (*Figure 1a*).

The L01 and L23 loops encircle the cavity forming the NA enzyme active site that binds sialic acid. NA cleaves off host sialic acids, aiding the release of newly formed virions from infected cells (*Palese et al., 1974*). While the loop residues contributing to the active site of NA are completely conserved, the remaining sequence of these loops can be highly variable. Many antibodies that inhibit enzyme activity and are protective in vivo bind to these loops (*Colman et al., 1983*; *Jiang et al., 2020*; *Yasuhara et al., 2022*; *Momont et al., 2023*; *Stadlbauer et al., 2019*; reviewed in *Lu et al., 2014*, *Figure 1—figure supplement 2*, *Figure 1—figure supplement 3*). Multiple studies have also shown that the majority of, but not all (*Figure 1—figure supplement 2*), monoclonal antibodies (mAbs) to NA select resistant viruses with amino acid replacements in the L01 and L23 loops. Analysis of the evolution of NAs has shown that the great majority of sequence change over time occurs at the surface exposed residues of NA, which include the residues of the L01 and L23 loops (*Colman et al., 1983*; *Bhatt et al., 2011*).

We assumed that the interactions between monomers in the NA tetramer are likely to make a major contribution to the stability and efficiency of expression of recombinant tetrameric NA in vitro. From known crystal structures, *Ellis et al., 2022* identified 44 residues that may contribute to these contacts in the N1 protein, 10 of which occurred in loops L01 and L23 (as defined by *Varghese et al., 1983*). In their final design of the stabilised sNAp-155 N1 2009 NA, they replaced 10 of these 44 amino acids, of which only 2 residues (conserved in N1 NA) were in the surface loops B2L01 and B2L23. Thus, the majority of intermonomer contacts are *not* in the L01 and L23 surface loops. In addition, the tertiary structure of NA is highly conserved. Within an NA subtype, the Cα atoms of the NA head (including surface loops) are superimposable with average root mean square deviation (RMSD) between the Cα atoms of less than 0.37 Å for N1 (*Xu et al., 2008*; *Sun et al., 2014*). From these observations, we reasoned that it might be possible to combine the antigenic features of a poorly expressed NA with the expression and stability properties of a high-expressing NA by grafting the L01 and L23 loops from the former onto the structural 'scaffold' of the latter, particularly within an NA subtype.

## Results
### Design of loop-grafted NA hybrid proteins

Using the PDB 1NN2 (H2N2 A/Tokyo/3/1967) structure and the top surface loop annotations by *Varghese et al., 1983* as a guide, we annotated the twelve L01 and L23 loops of the N1 NAs (*Figures 1a, b and 2*, *Figure 2—figure supplement 1*, *Supplementary file 2*). We transferred 12 dissimilar residues within the loops of N1/09 and 16 dissimilar residues of N1/19 to the H5N1/2021

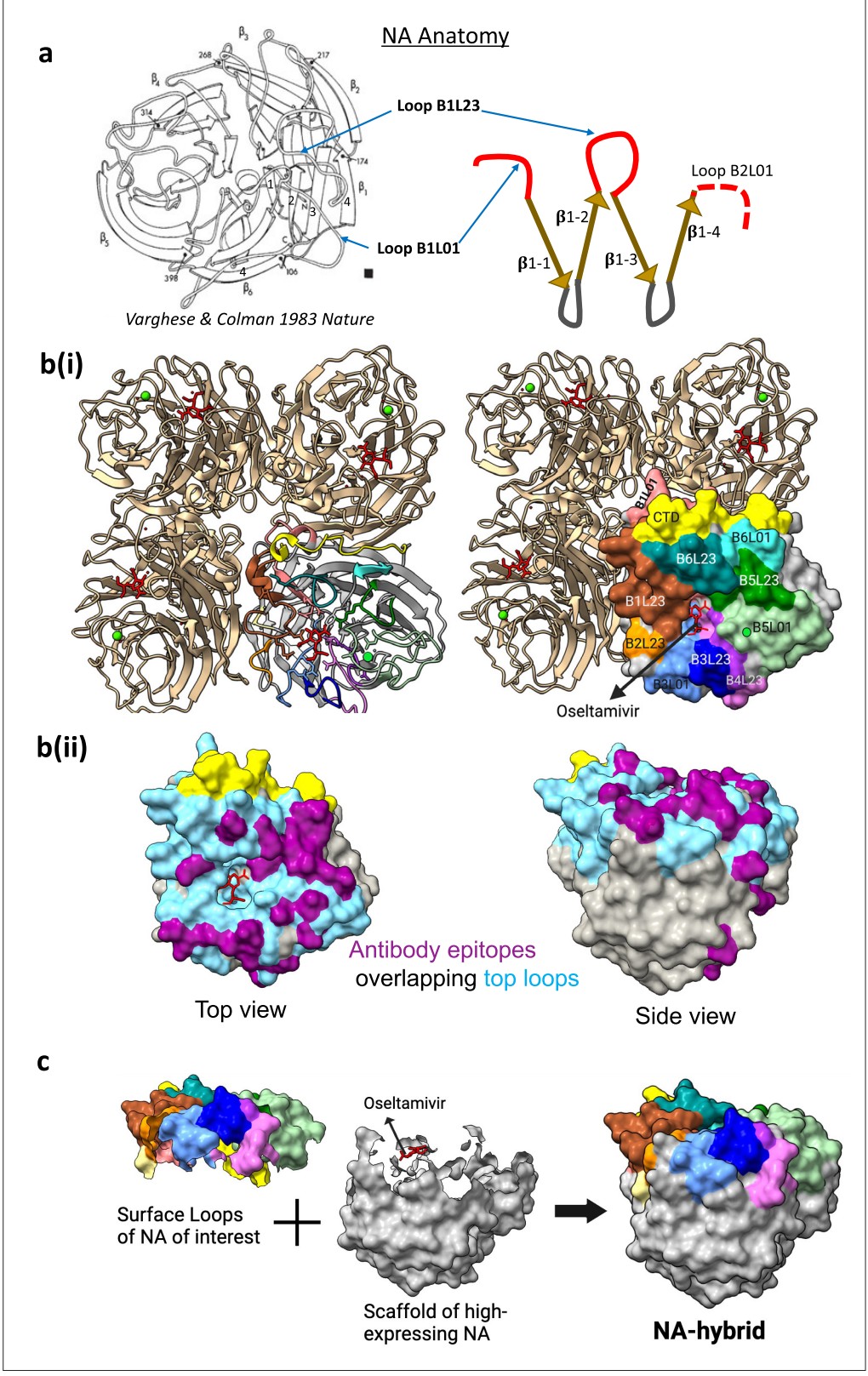

**Figure 1.** Surface antigenic loops transfer method to design NA hybrid protein. (**a**) NA anatomy. Each monomer of the tetrameric NA head is composed of a six-bladed propeller-like structure. Each blade unit consists of a β-sheet composed of four β-strands connected by loops arranged in a W-shape. The loop preceding β-strand 1 is termed loop 01 and the loop connecting β-strands 2 and 3 is termed loop 23. The twelve 01 and 23 loops in each

*Figure 1 continued*

monomer point up and surround and contribute to the enzyme active site. (**b**) Antigenic surface formation. (**i**) The key antigenic surface on the top of each NA monomer is formed by the twelve 01 and 23 loops from the six β-sheet units and the C-terminal domain. The 12 loops and the C-terminal domain in 1 monomer are coloured to show the top surface surrounding the active site. The remaining four β-strands and the loops 12 and 34 are referred to as the 'scaffold'. Oseltamivir is shown in red in the sialic acid receptor-binding site. A calcium ion at its binding site is shown as a green circle. (**ii**) Twelve L01 and L23 loops are coloured cyan blue and antibody epitopes as purple (L01 and L23 loops represent 90% of antibody epitopes – refer to *Figure 1—figure supplement 2* for epitope mapping on the aligned sequences). Similar to b(i), oseltamivir and C-terminal domain are shown for orientation. (**c**) Hybrid NA design. Loops 01 and 23, and the C-terminal domain and the scaffold are shown separately. Oseltamivir and the binding residues are included for reference. The surface antigenic loops 01 and 23 of an NA of interest were transferred on to the scaffold of a high-expressing NA candidate to improve the protein yield and stability. Figures were generated with PDB 4B7J using UCSF ChimeraX (*Meng et al., 2023*).

The online version of this article includes the following figure supplement(s) for figure 1:

**Figure supplement 1.** Graphical abstract.

**Figure supplement 2.** Viral neuraminidases selected for resistance to various monoclonal antibodies and sera with the positions of substitutions shown from seven published studies N1 numbering with H5N1 A/mute swan/England/053054/2021 (mSN1 as reference).

**Figure supplement 3.** Sequence alignment of various neuraminidases showing the contact residues for various mAbs, as defined by cryo-EM or crystal structures.

---

scaffold (mSN1) to create N1/09 and N1/19 hybrids (*Figure 2a, b*). From now on, the protein containing the N1/09 loops on the mS scaffold and N1/19 loops on the mS scaffold will be referred to as the N1/09 hybrid and N1/19 hybrid, respectively. The CTD of the final ~17 amino acids of the N1 protein also forms part of the top surface. However, in these N1 NAs, the CTD was conserved.

NA proteins were expressed in a transient mammalian ExpiCHO expression system using a gene construct comprising the N-terminus signal sequence, purification tags, SpyTag, a tetrabrachion (TB) tetramerisation domain (*Streltsov et al., 2019*) and the NA head ectodomain (*Figure 3—figure supplement 1a*, see Methods). The protein was purified from clarified supernatants and analysed using size-exclusion chromatography (SEC), SDS–PAGE with BS3 cross-linking, thermal unfolding, and enzymatic assays.

## N1/09 and N1/19 hybrid proteins expressed at higher levels as stable tetramers

We found that H5N1 A/mute swan/England/053054/2021 NA (mSN1) expressed at ~380 mg/l as a tetramer with minimal aggregation, exhibiting high enzymatic activity and a high melting temperature (*Figure 3a*). However, H1N1 A/California/07/2009 NA (N1/09) expressed at ~15 mg/l with a tendency to form aggregates, and H1N1 A/Wisconsin/588/2019 NA (N1/19) had a variable expression from undetectable to low (performed twice).

There was a substantial increase in protein yield for both N1/09 and N1/19 hybrid proteins (*Figure 3a*). These hybrids expressed as tetramers with negligible aggregation, as demonstrated by SEC (*Figure 3a*) and a BS3 cross-linked tetrameric complex of ~250 kDa resolved on SDS–PAGE (*Figure 3—figure supplement 1b*). The hybrids were enzymatically active in both small substrate MUNANA (580 Da) and large substrate fetuin (49 kDa) based assays. The N1/09 hybrid showed an 8°C improvement in its melting temperature $T_m$ to 64.4°C compared to the N1/09 protein (56.4°C), while the $T_m$ of the N1/19 hybrid (72.4°C) surpassed the H5N1 mSN1 scaffold donor (68.7°C). The 4.3°C difference in melting temperature between the mSN1 protein (68.7°C) and the N1/09 hybrid (64.4°C) may indicate that the sequence variation in the loops also contributes to stability of the tetramer.

Binding by mAb CD6 (*Ellis et al., 2022*; *Wan et al., 2015*) was also informative as CD6 binds across two protomers of the tetrameric NA and makes contacts with both loop and scaffold residues (*Figure 3—figure supplement 2a*). Ellis et al. have shown that binding of CD6 is dependent on the formation of compact closed conformation tetramers, whereas a more open conformation N1 2009 tetramer failed to bind CD6 (*Wan et al., 2015*).

Further, we compared the binding of CD6 to N1/09-VASP (with human vasodilator stimulated phosphoprotein tetramerisation domain) and N1/09-TB (with TB tetramerisation domain) proteins

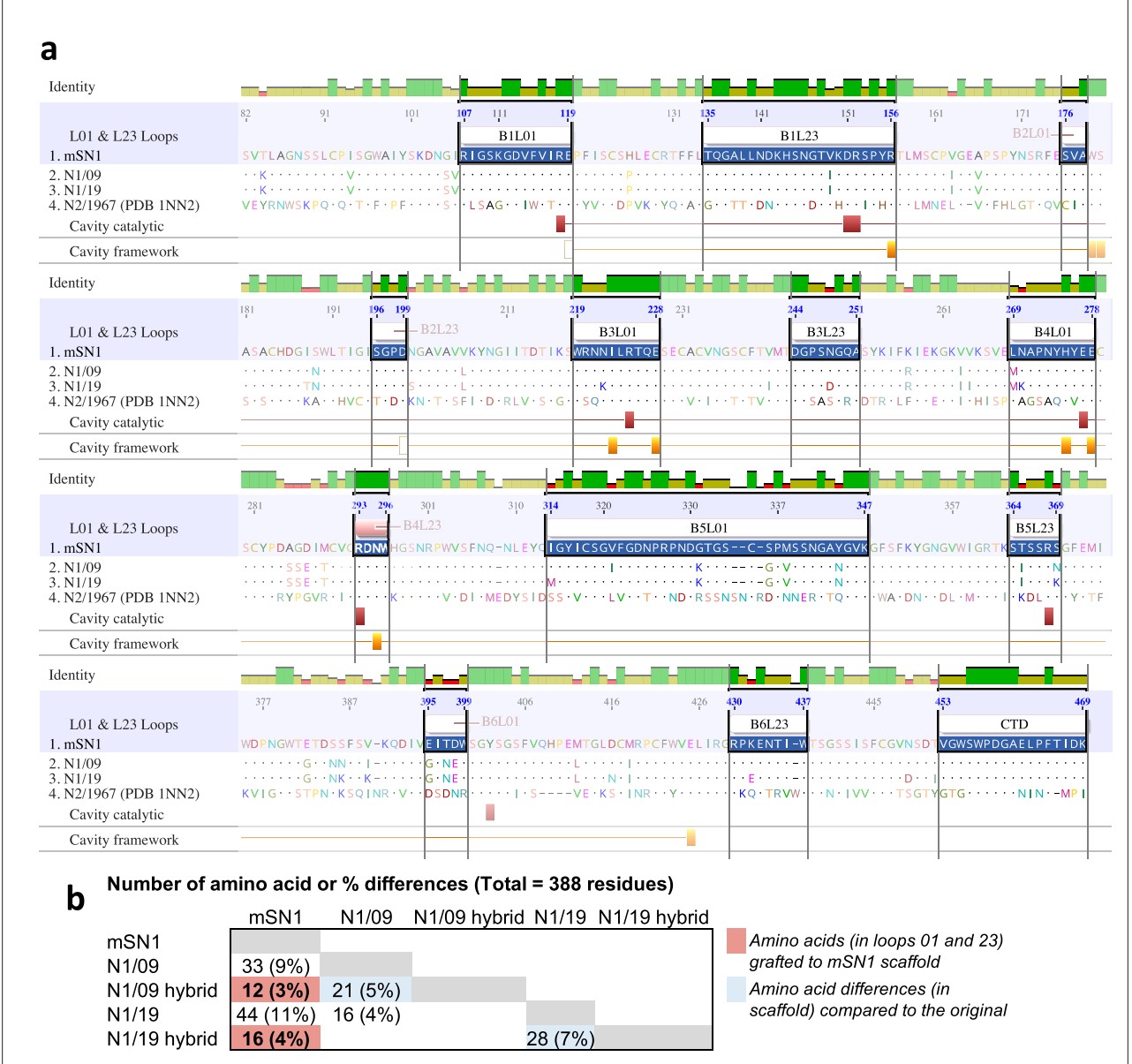

**Figure 2.** NA protein sequence alignment highlighting surface loops and active site. (**a**) Protein sequence alignment of the scaffold of N1 from H5N1 A/mute swan/England/053054/2021 (mS), and loop donors H1N1 A/California/7/2009 (N1/09) and A/Wisconsin/588/2019 (N1/19) neuraminidases. The N2 sequence H2N2 A/Tokyo/3/1967 from *Varghese et al., 1983*; *Varghese and Colman, 1991*, that was used to define the loops (PDB 1NN2) is also included. mSN1 is used as a reference sequence and identical residues are shown as dots. The sequence conservation is shown by green bars. The numbering here is based on N1 numbering as used in the annual Crick Reports. Loops 01 and 23 that form the top antigenic surface are highlighted. Loops annotation is based on Varghese et al. (*Supplementary file 2* and *Figure 2—figure supplement 1* for detailed information). Residues that form the catalytic site (8 residues) and conserved framework residues for the catalytic site (11 residues) of the enzymatic cavity are annotated with red and orange bars respectively. These residues are highly conserved between NA subtypes and the majority are part of the surface loops - 7/8 catalytic residues and 8/11 catalytic site framework residues. Two catalytic site framework residues are at the edge of loop B2L01. The figure was generated using Geneious Prime. (**b**) Number of amino acid differences for the N1/09 and N1/19 loop donors and the mSN1 loop recipient are shown. Twelve dissimilar residues within Loops 01 and 23 of N1/09 were transferred to mSN1 scaffold to form N1/09 hybrid. Similarly, 16 dissimilar residues within Loops 01 and 23 of N1/19 were transferred to mS N1 scaffold to form N1/19 hybrid.

The online version of this article includes the following figure supplement(s) for figure 2:

**Figure supplement 1.** Top loops 01 and 23 of neuraminidase, as defined by the first NA structure publications.

**Figure supplement 2.** Comparison with the optimised N1 sequence by Ellis et al.

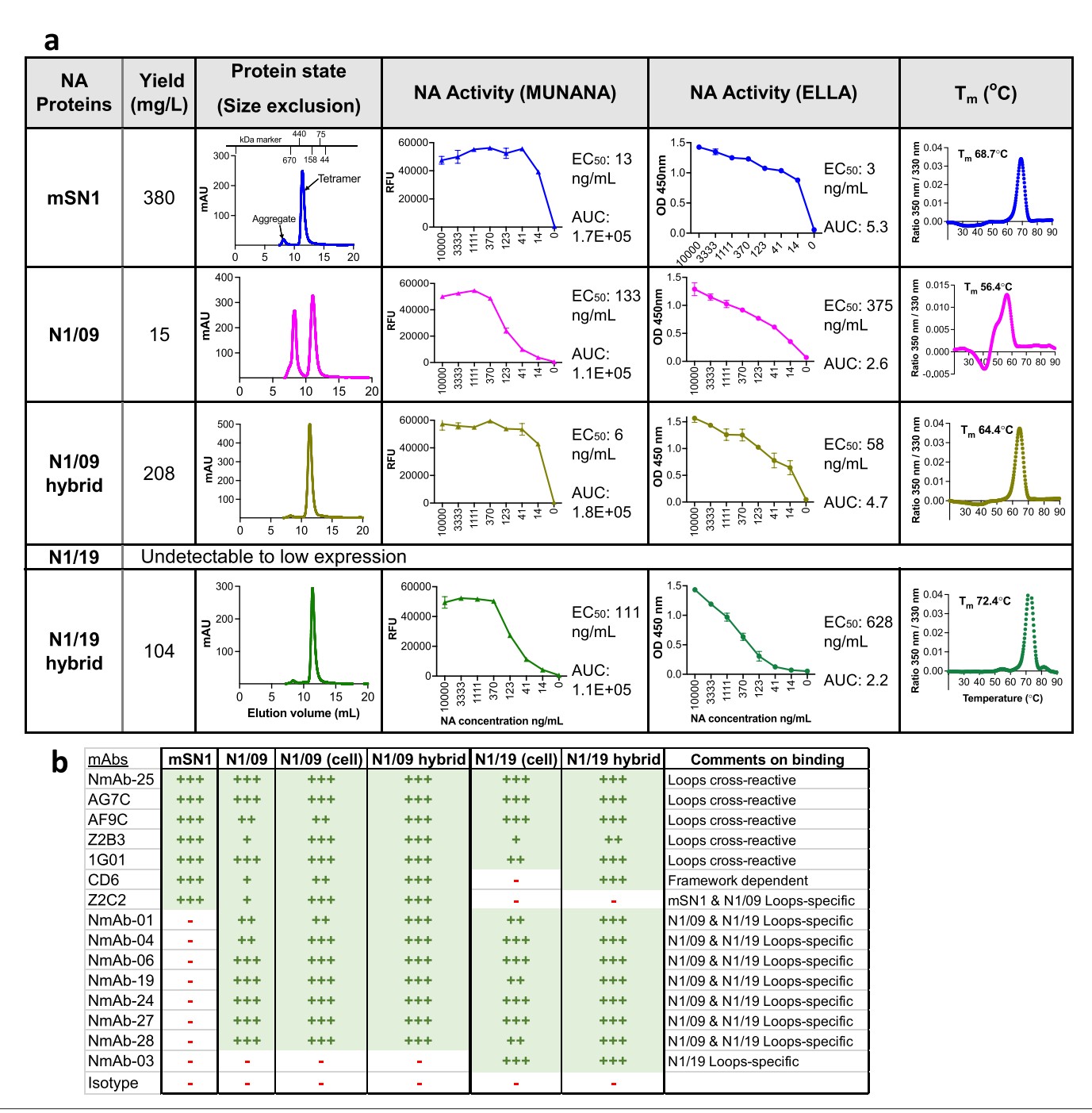

**Figure 3.** Characteristics of NA hybrid proteins including epitope specificity. (**a**) Characteristics of NA proteins. NA proteins were expressed in a transient mammalian ExpiCHO expression system. Gene constructs including affinity purification tags, SpyTag and an artificial tetramerisation domain tetrabrachion at N-terminus were cloned in the pcDNA3.1- vector. Proteins were purified using the 6His tag and Nickel-sepharose HisTrap purification columns. See *Figure 3—figure supplement 1* for constructs design and SDS–PAGE of purified proteins. N1/09 hybrid means residues of loops 01 and 23 of N1/2009 grafted to mSN1 (H5N1/2021) scaffold. N1/19 hybrid means loops 01 and 23 of N1/2019 grafted to mSN1 (H5N1/2021) scaffold. The expression yield of NA proteins and their hybrid forms is included in the second column. N1/19 protein expression was undetectable to low in ExpiCHO cells in two instances. Size-exclusion chromatography graphs are shown in the third column. Elution volume of 10–14 ml indicates tetrameric form of the protein. MUNANA and enzyme-linked lectin assay (ELLA) activity of the NA proteins are in fourth and fifth columns with $EC_{50}$ (effective concentration 50%) and AUC (area under curve) values, and the nanoDSF thermal melting temperature is in the final column. The sharp narrow peak and the higher melting temperature indicate higher protein stability. (**b**) Epitope specificity has been transferred with loops. Human monoclonal antibodies, previously published and some new, were titrated for ELISA binding of NA proteins. Areas under the curve were ranked after normalisation with one

*Figure 3 continued on next page*

*Figure 3 continued*

of the strongest binding mAb (see *Figure 3—figure supplement 3*) '+++' denotes >70% binding, '++' 40–70% binding, '+' 10–40%, and '-' <10% as a non-binder. Loops cross-reactive mAbs AG7C, AF9C, Z2B3, and 1G01, all defined by crystal structures, show full binding to all NA proteins. Seven mAbs (NmAb) do not bind mSN1 but bind to N1/09, N1/19 and their hybrid forms. NmAb-03 is specific to N1/19 surface loops. mAb CD6 is a scaffold-dependent mAb that shows binding to N1/19 hybrid protein but does not bind the N1/19 protein on the cell surface. mAb Z2C2 is specific for mSN1 and N1/09, hence did not bind N1/19 or its hybrid form.

The online version of this article includes the following source data and figure supplement(s) for figure 3:

**Source data 1.** SEC, ELLA, MUNANA, and Tm data for *Figures 3 and 6*.

**Source data 2.** mAbs binding data related to *Figures 3b and 6c*.

**Figure supplement 1.** Neuraminidase expression constructs design and protein characterisation.

**Figure supplement 1—source data 1.** PDF file containing reducing SDS–PAGE for *Figure 3—figure supplement 1*, indicating the relevant lanes.

**Figure supplement 1—source data 2.** PDF file containing BS3-cross-linked reducing SDS–PAGE for *Figure 3—figure supplement 1*, indicating the relevant lanes.

**Figure supplement 1—source data 3.** Original files for SDS–PAGE displayed in *Figure 3—figure supplement 1*.

**Figure supplement 2.** Epitope and binding pattern of mAb CD6.

**Figure supplement 3.** Binding titration of mAbs against recombinant soluble proteins or NA expressed on the surface of virus infected cells.

expressed in ExpiCHO cells and found that CD6 recognised only the latter (*Figure 3—figure supplement 2b*). CD6 bound better to N1/09 infected cells than to the recombinant N1/09-TB protein (*Figure 3b*, *Figure 3—figure supplement 3a*). This aligns with Ellis et al.'s findings that the 2009 N1 protein (with a VASP tetramerisation domain) predominantly formed open tetramers that failed to bind CD6. Our recombinant N1/09 protein with the TB tetramerisation domain may have formed a mixture of open and closed tetramers, also suggested by the appearance of aggregates on SEC and the relatively low $T_m$ of 56.4°C (*Figure 3a*). In contrast, CD6 bound strongly to the N1/09 hybrid protein, which had a higher $T_m$ of 64.4°C. CD6 also bound strongly to mSN1 scaffold donor.

The CD6 epitope was lost on seasonal H1N1 viruses isolated after 2015 (unpublished observation) so CD6 bound to cells infected with N1/09 but not N1/19 virus, but CD6 bound to the N1/19 hybrid protein strongly, suggesting that CD6's binding to the N1/19 hybrid depended on the scaffold residues donated by the msN1 protein, and that the N1/19 hybrid protein had formed a compact tetramer compatible with the N1/19 hybrid's higher $T_m$ of 72.4°C (*Figure 3a*).

## N1/09 and N1/19 hybrid proteins retained epitope specificity and enzyme activity

Next, we assessed the NA hybrid proteins for the epitopes recognised by monoclonal antibodies isolated from humans infected with influenza virus or vaccinated with inactivated seasonal influenza vaccine. A comprehensive analysis was done using a collection of 25 human mAbs, isolated by us or synthesized in the laboratory based on published sequences (see Methods; *Figure 3—figure supplement 3a*). Results from the binding titration of 15 mAbs on mSN1 and hybrid proteins and NA on the virus-infected cell membrane are depicted in *Figure 3b*. These included a set of five broadly reactive mAbs, including four, 1G01 (*Stadlbauer et al., 2019*), Z2B3 (*Jiang et al., 2020*), AG7C, and AF9C (*Rijal et al., 2020* and personal communication with Dr. Yan Wu, Capital Medical University) with crystal structures showing binding to the L01 and L23 loops, and nine strain-specific mAbs that distinguished between loop donor and recipient. Cross-reactive mAbs showed full binding to all proteins – mSN1, loop-donors, and hybrid proteins. Seven mAbs which did not bind the mSN1 protein but bound N1/09, also bound to the N1/09 hybrid protein, showing their specificity for the transferred loops. Similarly, NmAb-03, displaying specific binding to N1/19 NA on infected cells, exclusively bound to the N1/19 hybrid protein. By contrast, mAb Z2C2, which is specific for mSN1 and N1/09, bound neither N1/19-infected cells nor the N1/19 hybrid protein.

These results showed that for a wide collection of cross-reactive and specific mAbs, isolated from influenza-infected or vaccinated humans, binding was retained on the hybrid proteins. Together, these results suggested that the tertiary structure of the loops had been retained after grafting onto the msN1 scaffold.

NA inhibiting drugs, oseltamivir and zanamivir, inhibited the NA activity of the hybrid proteins (*Figure 3—figure supplement 1c*). Similarly, mAb 1G01, a known catalytic site targeting mAb, also inhibited the activity of mSN1 and N1/09 hybrid but not N1/19 hybrid. This result was expected since the mAb 1G01 lost its inhibitory activity on 2019 H1N1 seasonal influenza (*Momont et al., 2023*) due to the substitution N222K (N1 numbering) located within the loop B3L01 (unpublished observation, *Figure 3*, *Figure 3—figure supplement 1*). While enzyme inhibition was lost, detectable binding of 1G01 was retained to N1/19 cells and N1/19 hybrid proteins (*Figure 3*, *Figure 3—figure supplement 3*).

## mSN1, N1/09 hybrid, and N1/19 hybrid crystal structures are nearly identical

To assess whether loop grafting influenced the local or overall structure of the NA, we determined high resolution crystal structures of mSN1, the N1/09 hybrid, and the N1/19 hybrid (*Figure 4*, *Supplementary file 3*). Overlay analysis revealed that N1/09 hybrid and N1/19 hybrid were nearly identical in structure to mSN1, with overall RMSDs of <0.25 Å across equivalent Cα atoms upon overlay (*Figure 4b*; *Meng et al., 2023*). This is consistent with previous findings, where N1 NAs within a subtype typically exhibit a root-mean square (RMS) deviation between 0.2 and 0.4 Å (*Xu et al., 2008*; *Sun et al., 2014*). All segments of the scaffold region in both NA hybrids superposed extremely well onto mSN1, with RMS deviation values of less than 0.5 Å between the most distantly aligned Cα residue pairs. Minor deviations were observed in the B1L23 (150-loop) and B6L23 (430-loop) regions bordering the active-site cavity, with the RMS deviation distances of up to ~1.0 Å between aligned Cα residue pairs. The remaining loops showed RMS deviation values of less than 0.5 Å.

While the scaffold is structurally conserved across the three resolved structures, each possesses unique active site conformations, which are primarily influenced by the L01 and L23 loops surrounding the active site cavity. Analysis of mSN1, N1/09 hybrid, and N1/19 hybrid structures revealed some variability in active site conformations, with the greatest variability existing between the cavities of two msN1 molecules within the asymmetric unit of the crystal, reflecting different trajectories of the B1L23 (150-loop) (*Figure 4c*). Additionally, we observed that the Y344N substitution in the N1/09 and N1/19 hybrids slightly widened the enzymatic cavity compared to msN1.

The structural and monoclonal antibody-binding data thus showed that the 12 or 16 substitutions in loops matching the 2009 and 2019 seasonal N1s grafted onto the avian N1 scaffold did not affect the protein fold. The NA hybrid proteins exhibited a high degree of structural conservation, consistent with all previously published crystal structures of N1 NA proteins (*Xu et al., 2008*; *Russell et al., 2006*).

## N1/09 and N1/19 hybrid proteins are immunogenic and elicit NA enzyme inhibiting antibody responses

We have previously shown that recombinant NA presented on a mi3 particle is strongly immunogenic at doses as low as 0.1 µg (*Rahikainen et al., 2021*), in contrast to 10- to 100-fold higher doses of free NA protein often described in the literature (*Giurgea et al., 2020*). Our recombinant NA proteins were coupled onto mi3 virus-like particles (NA-VLP) via the covalent bond between the SpyTag on the NA protein and SpyCatcher003 sequence on the mi3 protein (*Figure 3—figure supplement 1d*). Mice were immunised with 0.5 µg NA-VLP adjuvanted with 1:1 vol/vol AddaVax (squalene-based oil-in-water nano-emulsion). Intramuscular immunisations were administered twice at 3-week intervals, and sera were harvested 3 weeks post-booster dose to assess the antibody response. NA activity inhibition (NAI) IC$_{50}$ (inhibitory concentration 50%) titres were measured using a fetuin-based enzyme-linked lectin assay (ELLA).

All NA proteins were immunogenic in mice, generating binding and NAI antibodies to themselves (*Figure 5a*). mSN1 elicited NAI antibody titres against itself that cross-inhibited viral N1/09 NA. However, N1/09 protein did not induce a cross-reactive response against mSN1, although the N1/09 hybrid did elicit antibodies that inhibited mSN1. This difference may have been due to improved structural integrity of the N1/09 hybrid, or possibly due to induction of inhibitory antibodies to the mSN1 scaffold.

The mSN1 and the N1/19 hybrid both induced strong inhibitory antibodies to themselves, but the sera to mSN1 failed to cross-inhibit the enzyme activity of N1/19-like virus (*Figure 5—figure*

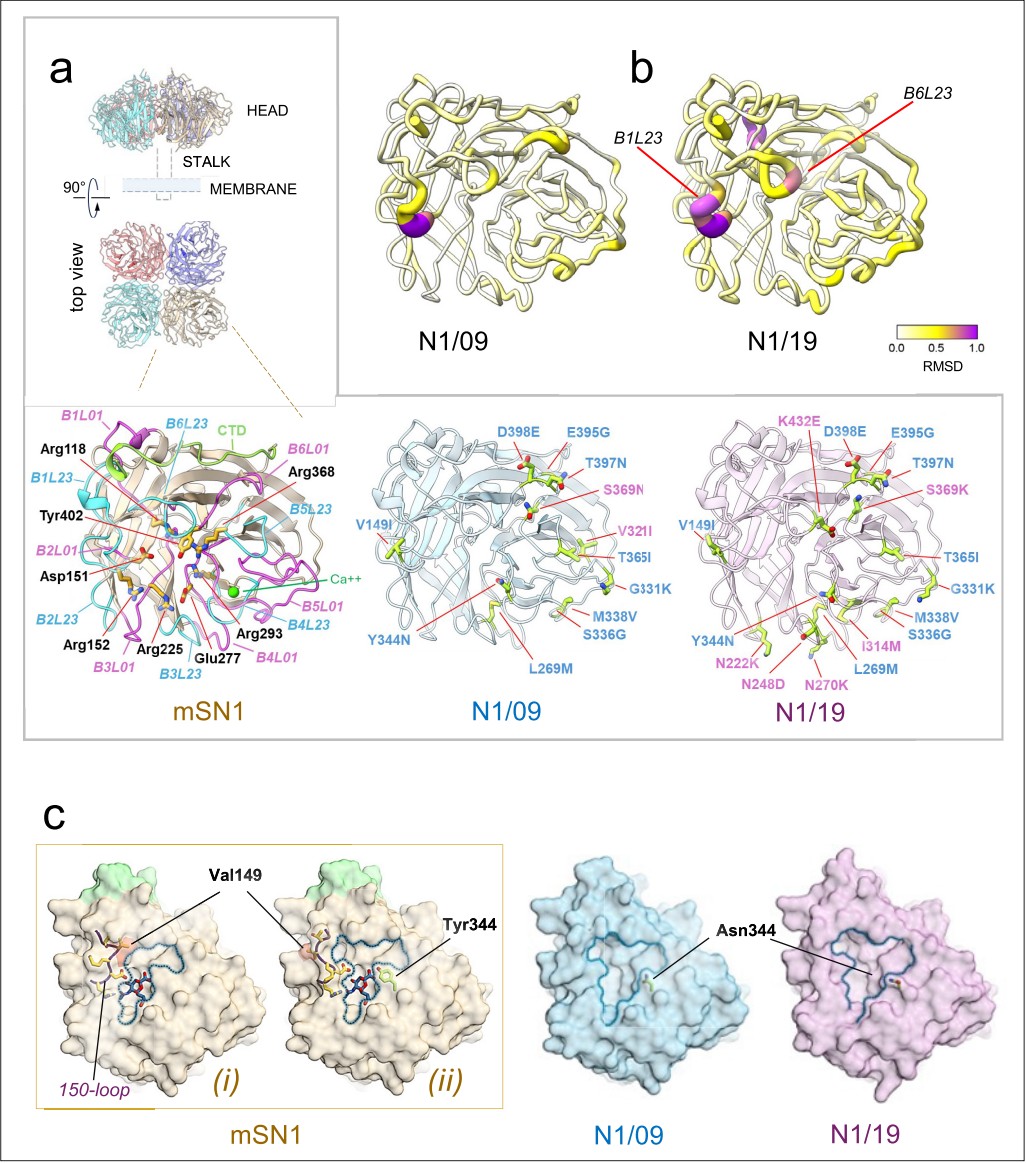

**Figure 4.** The structure of the influenza virus neuraminidase is strictly conserved in the loop-grafted hybrid proteins. (**a**) X-ray crystal structures. (*Top*) The influenza virus neuraminidase is shown in the native tetrameric form, an assembly observed in all reported crystal structures. A schematic representation of the stalk and a membrane is shown. (*Bottom*) Structures of each crystallized construct viewed from above the β-propeller fold (cartoon representation). (*Left*) Crystal structure of a protomer of the mSN1 neuraminidase. The β-strands of the H5N1/21 protomer (mSN1, left) are coloured pale brown, with the B6L01 and B6L23 loops coloured magenta and cyan blue, respectively. The C-terminal domain (CTD) is shown in pale green, and the calcium ion is depicted as a green sphere. Eight highly conserved residues in the catalytic site (R118, D151, R152, R225, E277, R293, R368, and Y402, N1 numbering based on msN1) are shown as yellow sticks. (*Middle*) Crystal structure of a protomer of the N1/09 loops-mS hybrid (pale blue). (*Right*) Crystal structure of a protomer of the N1/19 loops-mS hybrid (pale violet). B6L01 and B6L23 residues grafted into the N1/09 and N1/19 hybrid proteins are shown as yellow–green sticks. Residues identical in the N1/09 and N1/19 hybrids are labelled in blue; residues differing between these hybrids are labelled in violet. (**b**) Mapping structural differences between N1/09 and N1/19 hybrid structures. Local root-mean square (RMS) deviations between equivalent Cα pairs are mapped following the overlay of N1/09 (*left*) and N1/19 (*right*) onto the crystal structure of msN1. The b-propeller of msN1 is shown in a putty tube representation, with colour and radius reflecting the local RMS deviation values between equivalent Cα pairs. RMS deviations are generally below 0.5 Å, with modestly higher values in the inherently flexible B1L23 and B6L23 loops. (**c**) Active site cavities. The msN1 surface is coloured brown, except for the C-terminal loop region (green). The N1/09 and N1/19 surfaces are in blue and violet, respectively. In mSN1, the rim of the cavity containing the active site is traced

*Figure 4 continued on next page*

*Figure 4 continued*

with a blue dashed line. The position of the active site is denoted by a sialic acid molecule (grey–blue sticks), taken from a superposed, related structure (PDB ID: 2BAT). The msN1 structure revealed differences among the protomers of the tetramer at the active site entrance. Within the same tetramer, protomers with a relatively narrow cavity (i) combine with protomers showing a wider entrance (ii). This difference is dictated by the trajectory of the B1L23 loop (*150-loop*; dark blue with yellow side chains). The variation in trajectory between (i) and (ii) is most pronounced at Val149 (red surface). In contrast, the N1/09 and N1/19 hybrids show no noticeable differences between the protomer cavities, all of which closely resemble the wider msN1(ii) conformation, apart from minor, local widening of the rim induced by the substitution of Y344 with an asparagine residue (shown as light-green sticks).

---

*supplement 1*) and the N1/19 hybrid protein did not generate a cross-inhibitory NAI antibody response to mSN1. So, in this case the common mSN1 scaffold did not generate cross-inhibitory antibodies. N1/19 had four significant additional amino acid substitutions (see discussion) compared to N1/09 in the transferred L01 and L23 loops which may have been responsible for the loss of cross-reactivity between 2019 N1 with the 2021 avian mSN1. However, the N1/19 hybrid did generate antibodies that cross-inhibited N1/09 virus, so all four immunogens mSN1, N1/09, N1/09 hybrid, and N1/19 hybrid induced antibodies that could inhibit H1N1/2009 NA.

## Murine challenge studies

We challenged immunised DBA/2 mice with a lethal dose of H1N1/2009 (X-179A) virus and monitored weights for at least 14 days. Mice immunised with empty VLP lost ≥20% initial weight and hence were culled within 5–7 days post-infection. Vaccination with all the tested NA proteins mSN1, N1/09, N1/09 hybrid, and N1/19 hybrid protected mice from severe weight loss on challenge with X-179A. These results mirrored several studies in the literature which showed that immunisation with the 2009 N1 could provide at least partial protection in mice and ferrets to the avian H5N1 challenge (*Easterbrook et al., 2012*; *Sandbulte et al., 2007*; *Rockman et al., 2013*), and is compatible with the well charac-terised human mAbs that cross-inhibit broadly within the N1 subtype (*Rijal et al., 2020*, reviewed in *Rajendran et al., 2021*; *Zhang and Ross, 2024*).

In this case, all four immunogens induced inhibitory antibody to 2009 viral NA, resulting in protection from challenge with H1N1/2009 virus.

Although there is broad cross-reactivity of antisera within the N1 subtype (*Easterbrook et al., 2012*; *Sandbulte et al., 2007*; *Rockman et al., 2013*), reviewed in *Giurgea et al., 2020*, it is not universal (*Lu et al., 2014*; *Wu et al., 2012*) as demonstrated above with mSN1 and the N1/19 hybrid (*Figure 5—figure supplement 1*) which differed by 16 amino acids in the transferred loops, compared to 12 differences for the N1/09 hybrid (*Figure 2*). We noted that the Cambridge strain of H1N1 A/ PR/8/1934 and mSN1 differed by 18 residues in the L01 and L23 loops, which may be sufficient to prevent cross-inhibition. We then exchanged the L01 and L23 loops between A/PR/8/34 and mSN1 to look for correlation between antibody cross-reactivity and protection with the source of the L01 and L23 loops.

## Loop transfer between two distant N1 NAs: H5N1 A/mute swan/ England/053054/2021 (mS) and H1N1 A/PR/8/1934 Cambridge Strain (PR8)

mSN1 showed sufficient cross-reactivity to N1/09 to protect mice against virus challenge. Therefore, we performed loop transfer between mSN1 and PR8N1, which differ by 18 residues within the L01 and L23 loops and show no or minimal cross-reactivity, to assess the loop-specific protection. We exchanged the L01 and L23 top surface loops between mS and PR8 that differ by 18 residues (r of amino acid differences between mSN1 and *Figure 6a*, *Figure 6—figure supplement 1*). Following the nomenclature above, we will refer to these as the PR8 hybrid (PR8 Loops-mS) comprised of loops 01 and 23 from PR8 combined with the mSN1 scaffold and mS hybrid (mS loops-PR8) comprised of loops 01 and 23 from mS combined with the PR8 scaffold. Unlike the N1/09 and N1/19 proteins, wild-type PR8 N1 expressed well (~105 mg/l; $T_m$ 55.8°C) as tetramers with minimal aggregation (refer to SDS–PAGE in *Figure 3—figure supplement 1b*). The PR8 hybrid expressed at a nearly equal yield (108 mg/l; $T_m$ 58.1°C) and showed minimal aggregation (*Figure 6b*). Conversely, the mS hybrid gave a

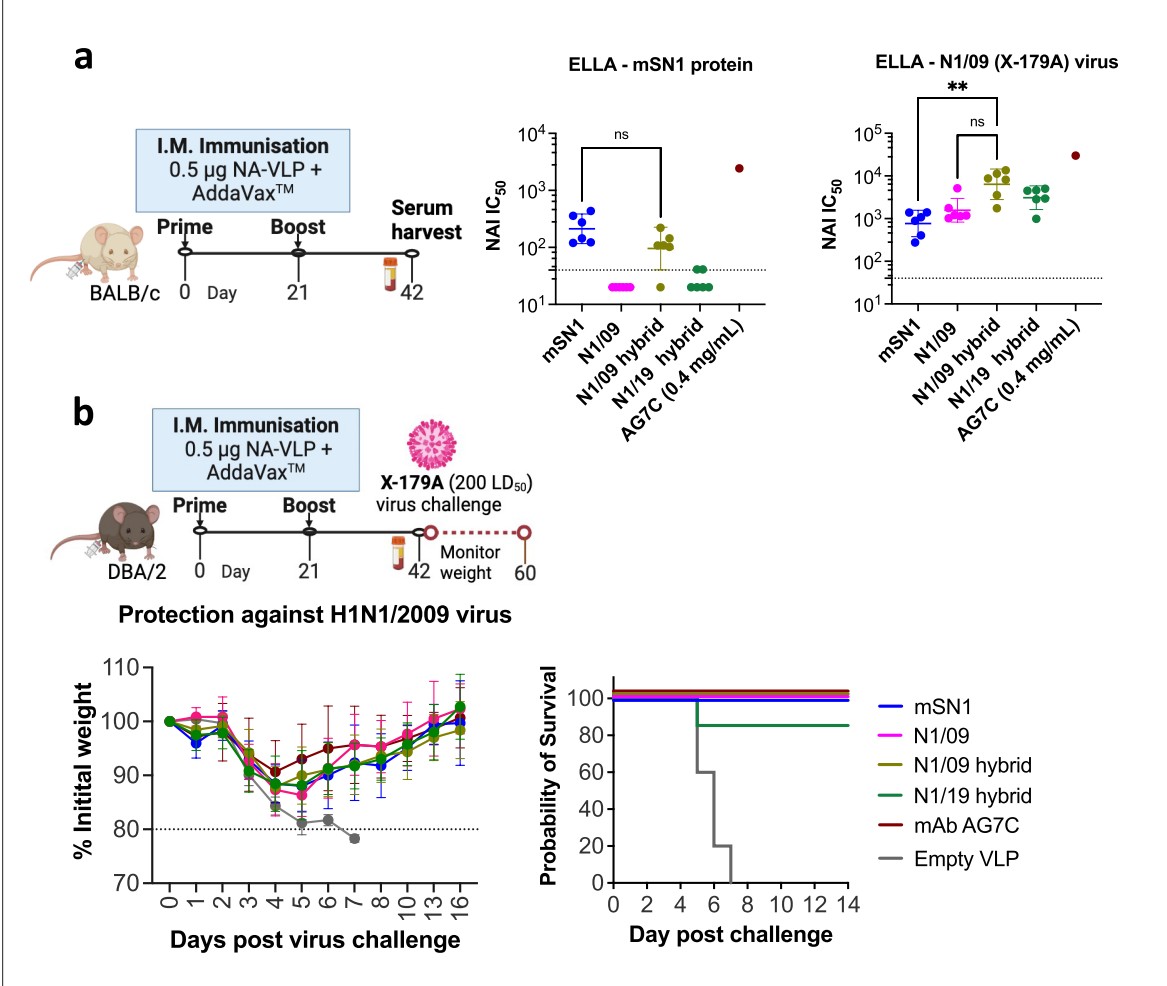

**Figure 5.** NA hybrid proteins are immunogenic and provide in vivo protection against virus challenge. (**a**) Immunogenicity of NA hybrid proteins. BALB/c mice ($n = 6$/group) were immunised with 0.5 µg NA coupled to the mi3 virus-like particles (NA-VLP) adjuvanted with 1:1 vol/vol AddaVax (squalene-based oil-in-water nano-emulsion). Intramuscular immunisations were done twice at the interval of 3 weeks and sera were harvested 3 weeks post booster dose to assess the antibody response. Neuraminidase activity inhibition (NAI) $IC_{50}$ titres measured using fetuin-based enzyme-linked lectin assay (ELLA) are shown as a separate dot for each mouse. mAb AG7C was used as a positive control. Pooled sera to unconjugated mi3 VLP was negative control and showed no inhibition at 1:40 dilution (not included here). Geometric mean with 95% confidence interval is shown. (**b**) Protection against virus challenge. DBA/2 mice ($n = 6$/group) were immunised as above and were challenged with intranasally administered 200 $LD_{50}$ of H1N1/2009 (X-179A) virus. Weights were monitored for 2 weeks. Loss of ≥20% initial weight was considered an endpoint. mAb AG7C (10 mg/kg prophylaxis) was used as a positive control. Empty VLP pooled sera was a negative control and mice reached the endpoint within day 5–7 post virus infection. The ELLA NA inhibition graphs of these pooled sera are shown in *Figure 5—figure supplement 1*. Figures were made using GraphPad Prism v10. Kruskal–Wallis test was used for statistical analysis. ns: non-significant (p-value >0.05), ** means p-value <0.005. Kaplan–Meier survival analysis was done with logrank Mantel–Cox test for comparison.

The online version of this article includes the following source data and figure supplement(s) for figure 5:

**Source data 1.** Mouse sera titration data on enzyme-linked lectin assay (ELLA).

**Source data 2.** Weight curves of mice following X-179A virus challenge.

**Figure supplement 1.** Inhibition of NA activity by mouse antisera measured using ELLA.

**Figure supplement 1—source data 1.** Mouse sera titration (pooled sera) on enzyme-linked lectin assay (ELLA).

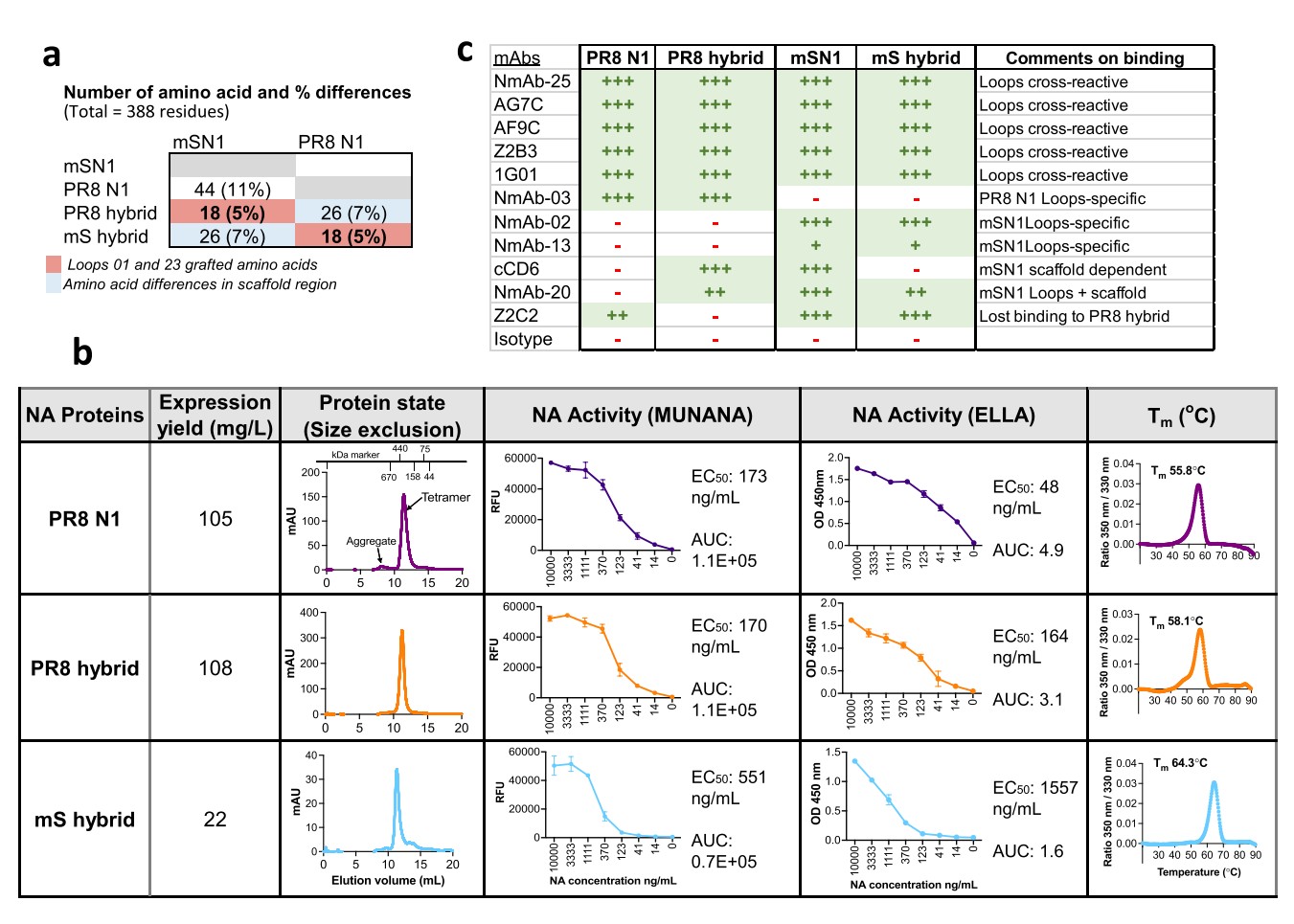

**Figure 6.** Loop grafting between two distant N1 NAs: H5N1 A/mute swan/England/053054/2021 (mS) and H1N1 A/PR/8/1934 (PR8). (**a**) Number of amino acid differences between mSN1 and PR8 N1 and their loop-grafted hybrids are shown. Eighteen dissimilar residues (5%) within loops 01 and 23 were grafted to make the hybrid proteins. (**b**) Characteristics of proteins. NA proteins were expressed in a transient mammalian ExpiCHO expression system. The expression yield of NA proteins and their hybrid forms is included in the second column. Size-exclusion chromatography graphs are in the third column. Elution volume of 10–14 ml indicates tetrameric form of the protein and 14–15 ml indicates trimeric or dimeric nature of the protein. MUNANA and enzyme-linked lectin assay (ELLA) activity of the NA proteins are in fourth columns and the nanoDSF thermal melting temperature is in the final column. The sharp narrow peak and the higher melting temperature indicate the higher protein stability. (**c**) Epitope specificity has been transferred with loops with a few exceptions. Antibodies, previously published and some new, were titrated for ELISA binding of NA proteins. Area under curve was ranked after normalisation with one of the strongest binding mAb (refer to *Figure 3—figure supplement 3b* for binding titration data). '+++' denotes >70% binding, '++' 40–70% binding, '+' 10–40%, and '-' <10% as a non-binder. Loops cross-reactive mAbs recognised both mSN1 and PR8N1 and their hybrid proteins. PR8N1 loops specificity is shown by NmAb-03. mAb CD6 is a major scaffold-dependent mAb (see *Figure 3—figure supplement 2*) and showed binding to PR8Loops-mS but not the PR8 and mSLoops-PR8. Similarly, NmAb-20 is a mSN1 Loops + scaffold-dependent mAb. mSN1 loops specificity is shown by NmAb-02 and NmAb-13.

The online version of this article includes the following figure supplement(s) for figure 6:

**Figure supplement 1.** Loop transfer between two distant N1 NAs: H5N1 A/mute swan/England/053054/2021 (mS) and H1N1 A/PR/8/1934 (PR8).

lower yield at (~22 mg/l; $T_m$ 64.3°C) but also showed a single peak in melting temperature and assembled into a tetramer with minimal aggregation (*Figure 6b*).

## Epitope specificity between loop-exchanged mS and PR8 hybrid proteins

We assessed the epitope specificity of these proteins, following the methods applied for N1/09 and N1/19 hybrids (*Figure 3—figure supplement 3*). We titrated 17 mAbs for binding to the wild-type and hybrid NA proteins, with results for representative 11 mAbs shown in *Figure 6c*. Ten cross-reactive

mAbs showed full binding to all proteins – PR8 N1 and mSN1 and their loops-exchanged hybrids. Among specific mAbs, NmAb-03 specifically bound to PR8 N1 and the PR8 hybrid but did not bind to mSN1 or the mS hybrid, thus showing specificity for the PR8 loops. Similarly, NmAb-02 and -13 bound to mSN1 and the mS hybrid but not to PR8 N1 or the PR8 hybrid, thus showing specificity for the mS loops. CD6 is a scaffold-dependent mSN1 binder and did not bind to PR8 N1 NA. CD6 maintained its binding to mSN1 scaffold and to the PR8 hybrid but not to the mS hybrid, indicating its dependence on scaffold residues from the H5N1 mS donor (*Figure 3—figure supplement 2a*).

mAbs NmAb-20 and Z2C2 were exceptions. NmAb-20 bound mSN1 and the mS hybrid, suggesting that it was mSN1 loop specific. However, in addition, it *gained* binding to the PR8 hybrid. mAb Z2C2 bound to both PR8 and mSN1, but lost binding to the PR8 hybrid, implying that this epitope had been lost in the PR8 hybrid protein. These two exceptions out of 17 mAbs studied in detail suggest that the transferred loops may show a limited amount of antigenic difference to either donor, perhaps at the margins of the loops.

## mS and PR8 hybrid proteins elicited loop-specific NAI antibodies and provided loop-specific protection in vivo against virus challenge

BALB/c mice were immunised with 0.5 µg NA-VLP adjuvanted with AddaVax (*Figure 7a*), as previously described for N1/09 in vivo experiments. mSN1 and mS hybrid proteins generated equivalent NAI sera titres towards mSN1, whereas PR8 N1 and the PR8 hybrid elicited no detectable titres to mSN1 (*Figure 7b*). Similarly, PR8 and PR8 hybrid generated equivalent titres against PR8 virus (*Figure 7c*). Interestingly, the mS hybrid elicited a low NAI titre (albeit ~10-fold lower compared to PR8 N1; p = 0.0009) against PR8 virus, implying that in this combination some inhibitory antibody may have been generated either to the PR8 scaffold or to conserved epitopes in the L01 and L23 loops.

In an independent experiment, we challenged immunised mice with a lethal dose of PR8 virus (Cambridge strain) (*Figure 7d*). Weight-survival curves showed that 6/6 mice immunised with PR8 NA and the PR8 hybrid NA survived without weight loss, whereas all mice vaccinated with the mS hybrid reached the endpoint, similar to the negative control group immunised with empty VLP (*Figure 7f, g*). For mice immunised with mSN1 wild-type NA, 2/6 mice survived but with significant weight loss. These experiments demonstrated that survival matched pre-exposure to vaccine presenting the PR8 L01 and L23 loops. The serology of vaccinated mice showed the same pattern (*Figure 7e*). PR8 and the PR8 hybrid vaccines generated inhibitory titres to PR8 virus NA, but mSN1 and the mS hybrid elicited no detectable NAI titres to PR8 virus (<160 in this experiment). The majority of the NAI serum titres were generated against the loops and this matched protection in this pair of NAs that differed by 18 residues in the L01 and L23 loops.

## Discussion

Our loop-grafting concept is straightforward and may be broadly applicable within other NA subtypes. The concept is based on three simple assumptions: (1) the majority of protective epitopes for the antibody response are present in the L01 and L23 loops on the top surface of the NA head as defined by Varghese and Colman (*Varghese et al., 1983*; *Varghese and Colman, 1991*), (2) the stability of the NA tetramer depends mainly on interactions between monomers encoded in the remainder of the NA sequence (the 'scaffold'), and (3) the structure of the loops will be maintained when grafted from one scaffold to another, at least within an NA subtype (*Musunuri et al., 2024*; *Correia et al., 2014*; *Schoeder et al., 2021*).

### The role of the tetramerisation domain

In preliminary experiments, we compared the VASP and TB tetramerisation domains for expression and immunisation with the 2009 N1 protein. We found that the VASP domain supported the formation of an NA tetramer, but the tetrameric protein did not bind the CD6 antibody that binds across two monomers (*Figure 3—figure supplement 2b*), confirming results of *Ellis et al., 2022*. By contrast, the N1/09-TB did bind CD6 and protected mice against matched viral challenge, as shown here. This result suggested that, while the VASP-linked 2009 N1 does form tetramers as defined by size-exclusion and cross-linking, the protein may not be in an optimal state for vaccination. We therefore have used the TB domain to form tetramers throughout this report.

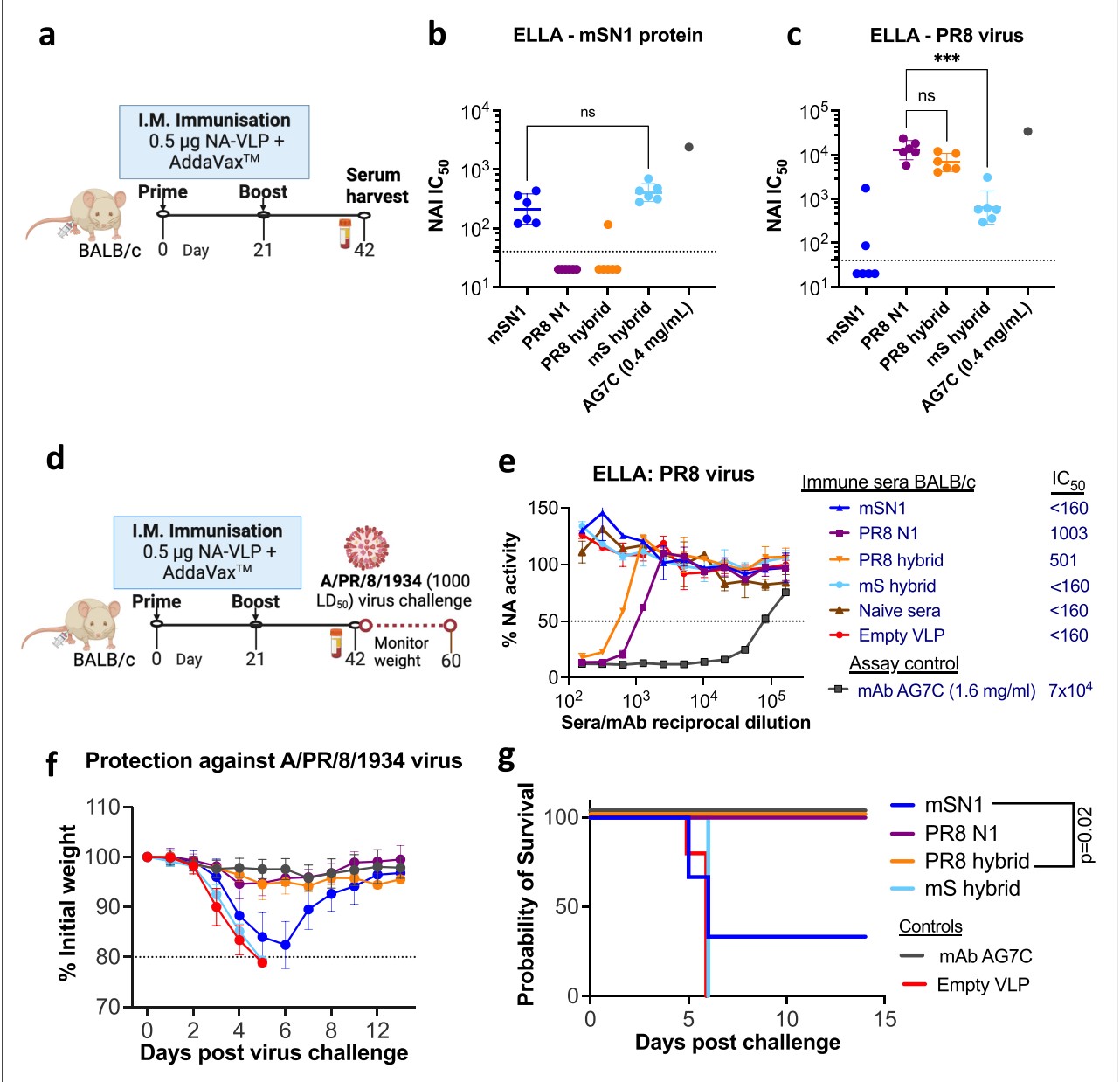

**Figure 7.** NA hybrid proteins elicited loop-specific NA inhibiting antibodies and provided loop-specific protection in vivo against virus challenge. (**a–c**) Immunogenicity of NA hybrid proteins. BALB/c mice (*n* = 6/group) were immunised with 0.5 µg NA coupled on to the mi3 virus-like particles (NA-VLP) adjuvanted with 1:1 vol/vol AddaVax (squalene-based oil-in-water nano-emulsion). Intramuscular immunisations were done twice at the interval of 3 weeks and sera were harvested 3 weeks post booster dose to assess the antibody response. Neuraminidase activity inhibition (NAI) $IC_{50}$ titres measured using fetuin-based enzyme-linked lectin assay (ELLA) are shown as a separate dot for each mouse. mAb AG7C was used as a positive control. Empty VLP pool sera were negative controls and showed no inhibition at 1:40 dilution (not included here). Geometric mean with 95% confidence interval is shown. (**d–g**) Protection against virus challenge. BALB/c mice (*n* = 6/group) were immunised as above. Pooled sera antibodies were assessed in ELLA assay before virus challenge (**e**) and $IC_{50}$ values shown are sera reciprocal dilution. Mice were challenged with intranasally administered 1000 $LD_{50}$ of PR8 virus (Cambridge strain, $10^4$ $TCID_{50}$). Weights were monitored for 2 weeks. Loss of ≥20% initial weight was considered an endpoint. mAb AG7C (10 mg/kg prophylaxis) was used as a positive control. Empty VLP pool sera were negative controls and mice reached the endpoint by day 6 post virus infection. Importantly, immunogens with PR8 Loops protected 100% mice from virus challenge, and mS loops did not. Mean and standard deviations are shown. Figures were made using GraphPad Prism v10. Kruskal–Wallis test was used for statistical analysis. ns: non-significant (p-value >0.05), \*\*\* denotes p-value <0.0005. Kaplan–Meier survival analysis was used with logrank Mantel–Cox test for comparison.

The online version of this article includes the following source data for figure 7:

**Source data 1.** Mouse sera titration data on enzyme-linked lectin assay (ELLA).

*Figure 7 continued on next page*

*Figure 7 continued*

**Source data 2.** Mouse sera titration (pooled sera) on enzyme-linked lectin assay (ELLA).

**Source data 3.** Weight curves of mice following PR8 virus challenge.

## The basis for stable tetramer formation

The hybrid proteins we produced all appeared to form stable tetramers as defined by SEC, BS3 cross-linking and melting temperature. They were all active in ELLA (large substrate) and MUNANA (small substrate) enzyme activity assays and were inhibited by standard small molecule inhibitors. We tested a wide range of virus-specific and cross-reactive human mAbs, and with few exceptions, the antibodies bound to the hybrid proteins. In addition, the CD6 antibody, that binds across two monomers and only to fully formed tetramers (*Wan et al., 2015*), bound to our hybrid proteins held together by the TB tetramerisation domain. Finally, the crystal structures of our hybrid N1 proteins confirmed that the grafted loops had retained their expected conformations. Together, these data suggested that the hybrid proteins had folded correctly and, in most pairs, combined the expression and stability characteristics of the scaffold donor with the antigenic properties of the loop donor.

Ellis et al. defined 44 residues that form contacts between N1 2009 monomers that could be involved in maintaining the stability of a tetramer (*Ellis et al., 2022*). In their final design of the stabilised sNAp-155 N1 2009 NA, they replaced 10 of these 44 amino acids to obtain a stable 2009 N1 tetramer supported by a VASP tetramerisation domain (*Figure 2—figure supplement 2*). In our hybrids, tetramerised with TB, these 10 residues are completely conserved between the scaffold and loop donors, so the source of stability for our hybrid proteins must lie elsewhere.

## The distribution of epitopes on NA

Varghese et al. described the structure of L01 and L23 loops on the top surface of N2 NA and showed that these loops contribute largely to the active site, notably including 7 of 8 conserved residues in the catalytic site and 8 of 11 conserved framework residues that support the site (*Varghese et al., 1983*; *Colman et al., 1983*). Also, in the L01 and L23 loops surrounding the active site, they noted multiple variable residues that vary seasonally associated with antigenic drift, and that can be selected for viral resistance by monoclonal antibodies in vitro. Resistance mutations selected for by three N2 murine mAbs (*Webster et al., 1982*) and N9 mAbs were within loops B3L01, B5L01, and B5L23 (*Colman et al., 1983*). Over the last 40 years, many similar studies have been done with murine and human monoclonal antibodies to various NAs. In *Figure 1—figure supplement 2*, we have collected a set that identifies 31 sites of amino acid substitution selected by monoclonal antibodies in independent experiments. Twenty-five of these are in loops 01 and 23, with three additional sites in immediate neighbour positions to these loops (90% altogether). Three additional sites of selection can be assigned to the underside of the NA head at the interface of stalk and head (position 88), B4 loop 12 (position 285), and B4 loop 34 (position 309). In addition, a recent study of human sera identified position 386 on B5 loop 34 as responsible for an antigenic change between N1/1977 and N1/1986 (*Daulagala et al., 2023*).

Crystal structures of bound Fab fragments give further information on the footprints of protective antibodies (*Figure 1—figure supplement 3*). The majority describe binding to the L01 and L23 loops, with recent examples also confirming binding to the side (CD6, NA-22) and underside of NA (*Wan et al., 2015*; *Yasuhara et al., 2019*; *Lederhofer et al., 2024*; *Zhu et al., 2019*; *Hansen et al., 2023*). It is important to note that loops 01 and 23 include a part of epitopes that have been described in the literature as side, lateral, or underside (see mAbs NDS.1, NDS.3, and CD6 in *Figure 1—figure supplement 3*). The majority of antibodies defined by crystallography bind within the surface of a single monomer, with recent structural evidence that a few mAbs bind across two monomers (*Wan et al., 2015*; *Zhu et al., 2019*). Finally, several structures of antibodies that bind within the active sites of a broad range of NAs have been described, all of which contact the L01 and L23 loops (*Jiang et al., 2020*; *Yasuhara et al., 2022*; *Momont et al., 2023*; *Stadlbauer et al., 2019*). From these data, we suggest that most antibodies generated by NA that are likely to be protective bind to the L01 and L23 loops, while a minority bind to epitopes on the underneath and side of the NA head that are not included within the L01 and L23 loops.

## Definition of loops

To align the top surface loops in N1 proteins, we relied on loop annotations from the N2 NA structure resolved to 2.9 Å by Varghese et al. as a reference (*Varghese et al., 1983*). The loop definitions in the structure later resolved to 2.2 Å differed slightly (*Varghese and Colman, 1991*; *Figure 2—figure supplement 1*). We suggest that loop annotations for a grafting experiment should not be rigid and require some judgement, particularly for residues at the margins of loops, taking into consideration solvent accessibility in crystal structures, evidence for evolutionary selection, viral resistance mutations selected with mAbs in vitro (*Figure 1—figure supplement 2*), and defined crystal structures of bound antibodies (*Figure 1—figure supplement 3*). For instance, in our N1/19 hybrid design, residue N200S could have been considered as part of loop B2L23 (*Varghese and Colman, 1991* and *Figure 2a*). In addition, the CTD contributes to surface residues and may be suitable for grafting, since residues within this section of NA have been selected by mAbs in vitro and clearly evolve over time (*Colman et al., 1983*; *Figure 1—figure supplement 2*, *Figure 1—figure supplement 3*). However, the CTD of N1 was conserved in all of our examples.

## Vaccination with hybrid NAs

Our NA vaccination strategy is based on our earlier evidence that linking tetrameric NAs to the mi3 vaccine-like particle (valency of 60) via a SpyTag/SpyCatcher covalent linkage results in enhanced immunogenicity and dose sparing, with doses as low as 0.1 µg of NA protein able to induce NAI antibody (*Rahikainen et al., 2021*). This compares well to the higher doses of pure protein used in the majority of studies of NA immunity (reviewed in *Zhang and Ross, 2024*). In the present experiments, we opted for two doses of 0.5 µg of mi3-linked NA that gave full protection in preliminary experiments (not shown), comparable to recent results from *Pascha et al., 2024*.

In our first set of grafting experiments between N1/2009 and mSN1 (H5N1 2021), we found that while protein expression was greatly improved, antisera from animals were cross-reactive for NA inhibition between the scaffold H5N1 donor, the hybrid (with 12 amino acid replacements in the L01 and L23 loops) and the 2009 loop donor. In addition, all immunised animals were protected from challenge with 2009 H1N1 virus. These results were consistent with evidence that within the N1 subtype, there is broad cross-reactivity for antibodies and that immunisation with 2009 N1 provides at least partial protection against an avian H5N1 (reviewed in *Zhang and Ross, 2024*). Cross-reactivity could have been for epitopes anywhere within the NA structure.

The sequence difference in the loops between 2019 N1 donor and the H5N1 recipient was greater (16 amino acids), sufficient to provide an antigenic distance that prevented cross inhibition by antisera from animals immunised with these two NAs. The NA inhibitory activity of the antisera was predominantly specific for the L01 and L23 loops. N1/19 had four significant additional amino acid substitutions compared to N1/09 in the transferred L01 and L23 loops: N222K (B3L01), N244D (B3L23), N270K (B4L01), and K432E (B6L23). All four of these residues have been selected for resistance by inhibitory mAbs in vitro (see *Figure 1—figure supplement 2*), or form contacts with mAbs demonstrated in crystal structures (*Figure 1—figure supplement 3*), so these residues are likely to have been responsible for the loss of cross-reactivity between N1/2019 and mSN1. We do not have a challenge model for the 2019 H1N1 virus, so we could not look for in vivo protection specific to the 2019 N1 loops.

The lack of cross-inhibition between sera raised to the avian mSN1 and the seasonal N1/19 showed that cross-reactivity of antisera to the N1 NAs is not universal and reflects the results of *Lu et al., 2014* who found minimal cross-reactivity of rabbit sera raised to a seasonal N1 from A/Beijing/262/1995 and an avian N1 from A/Hong Kong/ 483/1997 that we note differed by seventeen amino acids in the L01 and L23 loops.

We found that the NA of mouse-virulent H1N1 virus A/PR/8/1934 (Cambridge strain) differed by 18 amino acids in the L01 and L23 loops from the mS H5N1 scaffold donor. We therefore prepared two further hybrids in which the loops were exchanged between these two NAs. The relevant hybrids induced NA inhibitory sera that were largely loop specific, and protection against a high dose challenge with A/PR/8/34 was now dependent on matched L01 and L23 loops. In this case, antibodies to the scaffold could not have been protective, and protection correlated with the NA inhibition activity induced, mirroring the evidence from human challenge and vaccination studies (reviewed in *Giurgea et al., 2020*; *Zhang and Ross, 2024*).

Taken together, these results suggest caution should be exercised in designing vaccination strategies that rely on cross-protection between seasonal NAs and future pandemic NAs.

### Limitations of the study

The three assumptions listed at the start of the discussion may not always hold. Clear examples exist of protective monoclonal antibodies that bind to epitopes in the NA scaffold, including those that bind epitopes on the side (*Wan et al., 2015*; *Yasuhara et al., 2019*; *Zhu et al., 2019*) and underneath of the head domain (*Lederhofer et al., 2024*). If these formed the majority of the response to vaccination, our approach would fail due to the first assumption. However, we have found that at least for the N1 subtype, the transferred loops induce both NA inhibiting antibody and provide protection against viral challenge in the mouse. In addition, loops on the underside of NA could also be exchanged. The second assumption may also be sensitive to exceptions. Ellis et al. identified 44 residues that may contribute to the contacts between NA monomers, 10 of which we note are in loops L01 and L23 (*Ellis et al., 2022*). If the latter were variable and turned out to be critical for stability of the tetramer, our approach would fail. We note that one of the hybrids we produced (mS hybrid with PR8 scaffold) expressed at a lower level than either parent. Finally, the assumption that the tertiary structure of the loops is maintained after grafting may not completely hold as we have found occasional mAbs that alter their binding to the hybrid protein. However, the great majority of NA inhibitory and protective antibodies isolated from humans bound the four hybrid NAs we have made. In addition, crystal structures revealed that the transferred loops of 2009 and 2019 NA donors were virtually superposable on those of the H5N1 recipient.

While we acknowledge these potential shortcomings of our approach, we recommend the loop-grafting strategy as a simple and pragmatic method to improve the yield and stability of NA protein vaccine antigens. NA loop grafting can also serve as a useful starting point for analytical methods aimed at further improvement (*Ellis et al., 2022*; *Skarlupka et al., 2023*). Computationally optimised sequences (*Skarlupka et al., 2023*; *Job et al., 2018*) designed for cross-protection might also be improved by focusing attention on the variation within the L01 and L23 loops.

### Impact

Purified NA has been safely administered to humans as a vaccine candidate (*Kilbourne et al., 1995*) and NAI antibodies to NA have been shown to be an independent correlate of protection against influenza infection (reviewed in *Giurgea et al., 2020*). Enhancing current vaccines with NA proteins could substantially boost their effectiveness. A major challenge has been the production of NA at optimal yields. In this study, we developed an innovative solution via loop-grafting to produce stable and immunogenic NA protein at high yields that addresses this critical issue. Moreover, this study presents an epitope-based vaccine design and protein engineering for generating protective immunity, which could potentially be applied to other antigens.

## Materials and methods

**Key resources table**

| Reagent type (species) or resource | Designation | Source or reference | Identifiers | Additional information |
|---|---|---|---|---|
| Antibody | 1G01 (anti-NA human monoclonal) | *Chen et al., 2018* | PDB: 6Q23 | 10 µg/ml |
| Antibody | NmAbs, 24-1C (anti-NA human monoclonals) | In-house (Kuan-Ying Huang) | | 20 µg/ml |
| Antibody | Goat-anti-mouse immunoglobulins HRP (polyclonal) | Dako | P0447 | ELISA (1:800) |
| Antibody | Rabbit-anti-human IgG HRP (polyclonal) | Dako | P0214 | ELISA (1:1600) |
| Antibody | AG7C (anti-NA human monoclonal) | *Rijal et al., 2020* | | 20 µg/ml |
| Antibody | AF9C (anti-NA human monoclonal) | *Rijal et al., 2020* | | 20 µg/ml |
| Antibody | CD6 (anti-NA chimeric-human monoclonal) | *Wan et al., 2015*; this paper | PDB: 4QNP | 20 µg/ml |

*Continued on next page*

*Continued*

| Reagent type (species) or resource | Designation | Source or reference | Identifiers | Additional information |
|---|---|---|---|---|
| Antibody | Z2B3 (anti-NA human monoclonal) | *Rijal et al., 2020*; *Jiang et al., 2020* | PDB: 6LXI | 20 µg/ml |
| Cell line (*Canis lupus familiaris*) | MDCK-SIAT1 | ECACC | RRID:CVCL_Z936 | |
| Cell line (*Cricetulus griseus*) | ExpiCHO (Chinese hamster ovary) | Thermo Fisher | A29127; RRID:CVCL_5J31 | |
| Chemical compound, drug | *N*-tosyl-l-phenylalanine chloromethyl ketone (TPCK)-trypsin | Sigma-Aldrich | T1426 | 0.75–1 µg/ml |
| Chemical compound, drug | KPL SureBlue (TMB substrate) | SeraCare | 5120-0077 | |
| Chemical compound, drug | Pierce BS-3 | Thermo Fisher | A39266 | 6 mM |
| Chemical compound, drug | MUNANA [0-(4-methylumbelliferyl)541-a-D-*N*-acetylneuraminic acid] | Sigma-Aldrich | 69587 | 100 µM |
| Chemical compound, drug | AddaVax | InvivoGen | vac-adx-10 | 1:1 vol/vol |
| Chemical compound, drug | Oseltamivir | Sigma-Aldrich | TA9H9A9A73CA | 500 nM |
| Chemical compound, drug | Zanamivir | Sigma-Aldrich | SML0492 | 500 nM |
| Commercial assay or kit | QIAGEN Plasmid Maxi Kit | QIAGEN | 12162 | |
| Commercial assay or kit | HisTrap HP His tag protein purification columns (5 ml) | Cytiva | 17524801 | |
| Commercial assay or kit | Zeba Spin Desalting Columns, 7K MWCO | Thermo Fisher Scientific | 89889, 89891 | |
| Commercial assay or kit | Expifectamine CHO transfection kit | Thermo Fisher | A29129 | |
| Peptide, recombinant protein | Fetuin | Sigma-Aldrich | F3385 | 25 µg/ml |
| Peptide, recombinant protein | PNA–HRP (peanut agglutinin conjugated to horseradish peroxidase) | Sigma-Aldrich | L7759 | |
| Peptide, recombinant protein | SpyCatcher003-mi3 virus-like particles | Ingenza Ltd | Lot No. 108-23-001-007 | |
| Recombinant DNA reagent | pcDNA3.1- (plasmids) expressing NA | This paper | | Cloned between NotI and EcoRI sites |
| Strain, strain background (*Escherichia coli*) | DH5α Competent *E. coli* (High Efficiency) | NEB | C2987; RRID:CVCL_R748 | Electrocompetent cells |
| Strain, strain background (*Influenza A*) | A/PR/8/1934 (Cambridge) | WHO Influenza Centre (Crick, London) | | |
| Strain, strain background (*Influenza A*) | X-179A | NIBSC | 14/116 | |
| Strain, strain background (*Influenza A*) | A/Sydney/5/2021 | WHO Influenza Centre (Crick, London) | | |
| Strain, strain background (*Mus musculus*; female) | BALB/cOlaHsd | Envigo (inotiv) | Order code: 162 | |
| Strain, strain background (*Mus musculus*; female) | DBA/2OlaHsd | Envigo (inotiv) | Order code: 870 | |
| Commercial assay or kit | Morpheus II crystallisation screen | Molecular Dimensions | MD1-91 | 96-Reagent screen |
| Commercial assay or kit | Poly gamma glutamic acid crystallisation screen | Molecular Dimensions | MD1-51 | 96-Reagent screen |
| Commercial assay or kit | Pentaerytritol crystallisation screen | Jena Bioscience | CS-210L | 96-Reagent screen |
| Chemical compound, drug | Glycerol | Sigma-Aldrich | G7757 | |

*Continued on next page*

*Continued*

| Reagent type (species) or resource | Designation | Source or reference | Identifiers | Additional information |
|---|---|---|---|---|
| Software | Xia2 programme suite | *Winter et al., 2013* | | X-ray data processing software |
| Software | autoProc/StarAniso | *Vonrhein et al., 2018* | | X-ray data processing software |
| Software | Phaser | *McCoy et al., 2007* | | X-ray data processing software |
| Software | Coot | *Emsley et al., 2010* | | Model building and refinement software |
| Software | Phenix | *Liebschner et al., 2019* | | Model building and refinement software |
| Software | ChimeraX | *Meng et al., 2023* | | Software for structure visualization and analysis |

## Cell lines and viruses

ExpiCHO (Chinese hamster ovary) cells (Thermo Fisher) were used for the production of NA proteins and monoclonal antibodies. They were handled according to the manufacturer's protocol. MDCK-SIAT1 (Madin–Darby Canine Kidney cells stably transfected with human α 2,6- sialyltransferase, SIAT1) cells were used for the production of viruses, and for virus infection for epitope specificity assays (*Matrosovich et al., 2003*). The cell identity was confirmed by STR (short tandem repeat) profiling. Cells were maintained in D10 medium [Dulbecco's modified Eagle medium (DMEM) supplemented with 10% (vol/vol) foetal calf serum (Sigma-Aldrich, F9665), 2 mM glutamine, 100 U/ml penicillin, and 100 µg/ml streptomycin], with incubation in a humidified 5% $CO_2$ 37°C incubator. DMEM supplemented with 2 mM glutamine, 10 mM HEPES (*N*-2-hydroxyethylpiperazine-*N'*-2-ethanesulfonic acid), 0.1% BSA (bovine serum albumin), 100 U/ml penicillin, and 100 µg/ml streptomycin was used for virus growth and virus dilution. The medium is referred to as virus growth medium (VGM). All cell lines were tested to be mycoplasma-free.

Viruses were obtained from the Worldwide Influenza Centre, Crick Institute, UK and the National Institute for Biological Standards and Control, UK. Viruses were propagated by infecting a monolayer of MDCK-SIAT1 with ~0.1 MOI (multiplicity of infection) virus for 1 hr before replacing with VGM containing 0.5–1 µg/ml TPCK (L-1-tosylamido-2-phenylethyl chloromethyl ketone)-treated trypsin. Virus supernatant was harvested after 48 hr.

## Monoclonal antibodies

Antibodies AG7C, AF9C, Z2C2, and Z2B3 were previously published by our laboratory (*Rijal et al., 2020*). mAb 1G01 was produced in the laboratory using the antibody sequence obtained from PDB 6Q23 (*Stadlbauer et al., 2019*). Genes were synthesised by GeneArt (Life Technologies), cloned into antibody expressing vectors, and expressed in ExpiCHO cells. Similarly, mAb CD6 was synthesised using the sequence obtained from PDB 4QNP (*Wan et al., 2015*). The mAb is chimeric since the mouse variable region is cloned with the human constant regions.

NmAbs and 24-1C are anti-NA human mAbs provided by our collaborator Prof. Kuan Ying Huang (National University Taiwan), which we produced in ExpiCHO expression system using their expression plasmids (see *Figure 3—figure supplement 3*). The study protocol and informed consent of human monoclonal antibody isolation were approved by the ethics committee at the National Taiwan University Hospital and Chang Gung Memorial Hospital and were carried out in accordance with the Declaration of Helsinki and Good Clinical Practice guidelines. Written informed consent was received from each adult prior to inclusion in the study.

## NA protein expression and purification

Gene constructs (except for the N1/09 protein) were designed to have H7 HA signal sequence ( MNTQILVFALIAIIPTNADKI), Strep-tag II (SAWSHPQFEK), G3S linker, 6xHis, S3SG linker, SpyTag (AHIVMVDAYKPTK), G2SG4S linker, and TB tetramerisation domain from *Staphylothermus marinus* (*Streltsov et al., 2019*), GGSGTG linker, and finally an NA ectodomain (82–469) at the C-terminus

(*Figure 3—figure supplement 1a*). Sequences were synthesised as human-codon-optimized cDNAs by GeneArt (Life Technologies) and cloned into pcDNA3.1/- plasmid for transfection. Proteins were expressed in ExpiCHO expression system (Thermo Fisher). Briefly, ExpiCHO cells were cultured in a humidified Multitron Cell incubator (Infors HT) at 37°C with 8% (vol/vol) $CO_2$, rotating at 125 rpm, for at least two passages before transient transfection. The Max-titre protocol from the manufacturer was used and proteins were harvested on day 8/9 post-transfection. Culture supernatants were clarified by centrifugation at 3000 × *g* for 10 min at room temperature, followed by filtration through a 0.22-μm filter (RapidFlow Nalgene). Proteins were purified from cell culture supernatants using Ni-NTA-sepharose (prepacked HisTrap HP column from Cytiva) using the AKTA Pure purification system. The binding buffer was 20 mM sodium phosphate ($Na_2HPO_4$), 0.5 M NaCl, 20 mM Imidazole, pH 7.4, and the elution buffer was 20 mM $Na_2HPO_4$, 0.5 M NaCl, 500 mM imidazole, pH 7.4. The eluate was then buffer exchanged to Dulbecco's phosphate-buffered saline (DPBS) with calcium and magnesium (Gibco 14040133) using 7K MWCO (molecular weight cutoff) Zebaspin desalting columns (Thermo Fisher). Protein aliquots were stored at –80°C for long-term storage and at 4°C for short-term use and storage.

## Size-exclusion chromatography

The putative oligomeric state of NA proteins was assessed via SEC. Proteins were run on Superdex 200 Increase 10/300 GL column (Cytiva) equilibrated with PBS at a flow rate of 0.5 ml/min. Graphs were plotted using GraphPad Prism software.

## NA enzymatic activity assays

The enzymatic activity of NA proteins was measured using ELLA (large substrate fetuin Mw: 49 kDa) and MUNANA (20-(4-methylumbelliferyl)-a-D-*N*-acetylneuraminic acid; small fluorescent substrate Mw: 489 Da) assays.

### ELLA

ELLA was performed as previously described (*Rijal et al., 2020*; *Rahikainen et al., 2021*). In brief, serially diluted heat-inactivated sera were incubated with pre-titrated virus (NA source) or recombinant NA protein for 1 hr. The dilution medium was DMEM (Gibco) supplemented with 2 mM glutamine, 10 mM HEPES, 0.1% (wt/vol) BSA, 100 U/ml penicillin, and 100 μg/ml streptomycin. The virus or NA concentration was pre-determined by titrating to measure the lowest concentration at top plateau, ensuring at least a fivefold signal-to-noise ratio. The mixture was transferred to a Nunc Immuno-assay ELISA plate (Thermo Fisher, 439454) pre-coated overnight with 25 μg/ml fetuin (Sigma-Aldrich, F3385). The plate was incubated for 18–20 hr at 37°C with 5% (vol/vol) $CO_2$ in a tissue culture incubator. The NA enzymatic activity was detected by adding HRP (horseradish peroxidase)-conjugated peanut agglutinin (Sigma-Aldrich, L7759) at 1 μg/ml after washing the plates three times with PBS and then developing with 50 μl TMB (3,3',5,5'-tetramethylbenzidine) substrate (SeraCare). The enzymatic reaction was stopped after 5–10 min using 50 μl 1 M $H_2SO_4$ and the absorbance was read at 450 nm using a CLARIOstar plate reader (BMG Labtech).

### MUNANA

The protocol from the WHO Worldwide Influenza Centre, Crick Institute was adapted. The sialidase activity of NA proteins or viruses was measured by incubation of serially diluted NA (50 μl) in 32.5 mM MES (2-(*N*-morpholino)ethanesulfonic acid) buffer pH 6.5 containing 4 mM $CaCl_2$ with 50 μl of 100 μM MUNANA substrate. The reaction mix was incubated for 1 hr at 37°C and stopped by adding 50 μl stop solution (0.1 M glycine, 25% ethanol, pH 10.7). Changes in fluorescence were measured at 460 nm (excitation 355 nm) using a CLARIOstar. The NA concentration giving fluorescence intensity of 40–60 K units (~100-fold signal from background) was used in NA inhibition assays. NA inhibition titres of anti-sera or mAbs were determined by incubation of serially diluted sera/mAbs with NA/viruses for 1 hr before the addition of MUNANA substrate.

Area under curve and $EC_{50}$ for NA activity were measured using GraphPad Prism. % NA activity was calculated as $\{(X − Min)/(Max − Min)\} × 100$ where $X$ = measurement value, Min = buffer only, Max = NA or virus alone. The 50% inhibiting titre/concentration ($IC_{50}$) for NA inhibitors was determined using

non-linear regression curve fit on GraphPad Prism. All graphs were plotted using GraphPad Prism software.

## nanoDSF thermal unfolding

The thermal stability and unfolding of NA proteins were measured using the nanoDSF (differential scanning fluorimetry) technique on a Prometheus Panta instrument (Nanotemper). NanoDSF monitors changes in the intrinsic tryptophan or tyrosine fluorescence resulting from alterations in the 3D structure of proteins as a function of temperature. Capillaries were filled with 10 µl of NA protein at 0.4 mg/ml, placed into the sample holder and the temperature was increased from 20 to 90°C at a ramp rate of 1°C/min, with one fluorescence measurement taken per 0.1°C. The fluorescence intensity ratio ($Em_{350\,nm}/Em_{330\,nm}$) and its first derivative were calculated using the manufacturer's software. Two measurements were taken for each experiment, and the means are shown. The experiment was repeated to ensure reproducibility.

## Epitope specificity analysis

A collection of monoclonal antibodies developed in-house and published in the literature was used to determine their binding specificity to NA proteins. Antibodies were serially diluted fivefold from 10,000 to 1 ng/ml. mAbs were tested for binding to recombinant protein or NA expressed on the surface of MDCK-SIAT1 cells infected overnight with the virus. Antibody binding was detected using a secondary goat-anti human HRP antibody. The area under the curve was measured for each mAb's titration curve and normalised as a percentage of relative binding to one of the highest binding mAb for comparison. The binding experiments were repeated, and the data from a single experiment are included. Data analysis and graphs generation were performed using GraphPad Prism.

## mi3 VLP conjugations

SpyCatcher003-mi3 virus-like particles (Lot No. 108-23-001-007) produced in *Bacillus subtilis* were kindly provided by Ingenza Ltd. NA and VLP conjugations were done in DPBS with calcium and magnesium (Gibco 14040133) at various molar ratios. The conjugation was analysed using reduced SDS–PAGE. The conjugates were confirmed to have enzymatic activities and binding of mAbs equivalent to the unconjugated proteins.

## Mice immunisation and virus challenge

Animal experiments were conducted in compliance with the UK Animals (Scientific Procedures) Act Project License (PP9362617), following the principles of the 3Rs (Replacement, Reduction, and Refinement) and ARRIVE (Animal Research: Reporting of In Vivo Experiments) guidelines. See Appendix 1 for study design and power calculation. Female BALB/c OlaHsd or DBA/2 OlaHsd mice (7–8 weeks old at the start of the experiment) were obtained from Envigo and housed in individually vented cages in a specialised unit for infectious diseases. The mice were housed in accordance with the UK Home Office ethical and welfare guidelines, fed standard chow and had access to water ad libitum. Mice were anaesthetized with isoflurane (Abbott) and immunised via intramuscular injections with two doses of 0.5 µg of NA-VLP immunogens adjuvanted with AddaVax 1:1 vol/vol, administered 3 weeks apart. Serum samples were collected 3 weeks after the booster dose: intermediate samples were obtained via tail vein and terminal samples via cardiac puncture of euthanised mice. Whole blood collected in microtainer SST tubes (BD) was allowed to clot at room temperature for 1–2 hr before centrifugation at 10,000 × *g* for 10 min. The clarified sera were transferred to fresh tubes and stored at –20°C.

For virus challenge experiments, 50 µl virus (passaged and titrated in the laboratory) was administered intranasally to anaesthetised mice. Virus dose was $10^4$ $TCID_{50}$ (tissue culture infectious dose 50%) [200 $LD_{50}$ (lethal dose 50%)] of X-179A (H1N1 A/California/07/2009) in DBA/2 mice and $10^4$ $TCID_{50}$ (1000 $LD_{50}$) of H1N1 A/PR/8/1934 in BALB/c mice. The weight and clinical signs of mice were monitored regularly over a study period of 2–3 weeks until they either reached their endpoint (≤80% of their initial weight) or recovered to their initial weight. Mice reaching the endpoint were humanely euthanised. Kaplan–Meier survival analysis with the logrank Mantel–Cox test was used for comparison.

## Sequence alignment

NA sequences were obtained from PDB files where referred, or from Global Initiative on Sharing All Influenza Data (GISAID). Amino acid sequences of the NA were aligned using Muscle alignment in Geneious Prime, and all alignment figures were generated using Geneious Prime.

## Crystallisation, data collection, and structure determination

In each well of 96-well sitting drop plates (Greiner), 100 nl of purified NA was combined with 100 nl of precipitant. The concentrations used for crystallisation were 4.6 mg/ml for mSN1, 5 mg/ml for N1/09 hybrid, and 4.5 mg/ml for N1/19 hybrid. For crystal formation, the mixtures were equilibrated against 90 µl of precipitant at 20.5°C (*Walter et al., 2005*).

Crystals of msN1 were obtained using the Morpheus B2 precipitant [0.09 M halogens (NaFl, NaBr, and NaI), 0.1 M imidazole/MES pH 6.5, 30% ethylene glycol/PEG (polyethylene glycol) 8 K; Hampton Research]. The N1/09 hybrid crystallised using precipitant H4 of the Pentaerytritol screen (25% pentaerytritol ethoxylate 797, 0.1 M MES, 0.05 M MgCl$_2$, pH 6.5; Jena Bioscience) and the N1/19 hybrid using PGA (poly gamma glutamic acid)-screen condition G2 (5% PGA-low molecular weight, 30% PEG550 MME (monomethyl ether), 0.1 M Tris pH 7.8; Molecular Dimensions). Glycerol was added to 25% (vol/vol) for cryoprotection of the N1/19 hybrid crystals. Diamond beamline I24 (Harwell, UK) was used for diffraction data collection at 100 K. Data processing used the Xia2 programme suite (*Winter et al., 2013*) and, for the N1/19 hybrid, autoPROC and STARANISO (*Vonrhein et al., 2018*). The molecular replacement program PHASER (*McCoy et al., 2007*) was used for structure elucidation. Model building was done using COOT (*Emsley et al., 2010*) and model refinement using Phenix (*Liebschner et al., 2019*). Data collection and refinement statistics are provided in *Supplementary file 3*. Figures were prepared using UCSF ChimeraX (*Meng et al., 2023*).

## Materials availability

Custom antibodies and plasmids generated for this study are available from the corresponding authors upon reasonable request, subject to completion of a material transfer agreement if required.

## Acknowledgements

We thank Ingenza Ltd for providing SpyCatcher003-mi3 particles, and Dr Rodney Daniels and Dr John McCauley for providing influenza viruses. Thanks to Max Quastel, Dr Jack Tan, Lisa Schimanski, and Diana Melnyk for laboratory assistance. The authors would also like to thank Diamond Light Source (Harwell, UK) for beamtime (proposal mx28534), and the I24 beamline staff for assistance with data collection. This work was supported by the Chinese Academy of Medical Sciences (CAMS) Innovation Fund for Medical Science (CIFMS), China (grant number: 2024-I2M-2-001-1) to PR and AT, and the Townsend-Jeantet Prize Charitable Trust Charity No. 1011770. The work of monoclonal antibody development was supported by the National Science and Technology Council of Taiwan (NSTC 111-2628-B-002-053-, 112-2314-B-002-099-MY3), and the National Taiwan University (NTU-CC-113L892904, NTU-CC-114L891104) to K-YAH. The structural work was funded by the Medical Research Council MR/S007555/1 to TAB. The Centre for Human Genetics was supported by the Wellcome grant 203141/Z/16/Z.

## Additional information

### Competing interests

Pramila Rijal, Alain RM Townsend: is an inventor on patents filed on Influenza Neuraminidase loop-based design (GB application N430510GB). Mark Haworth: is an inventor on patents filed on spontaneous amide bond formation (EP2534484 and UK Intellectual Property Office 1706430.4) and a SpyBiotech co-founder and shareholder. The other authors declare that no competing interests exist.

## Funding

| Funder | Grant reference number | Author |
| --- | --- | --- |
| Chinese Academy of Medical Sciences | 2024-I2M-2-001-1 | Pramila Rijal<br>Alain RM Townsend |
| National Science and Technology Council | 111-2628-B-002-053- | Kuan-Ying A Huang |
| National Science and Technology Council | 112-2314-B-002-099-MY3 | Kuan-Ying A Huang |
| Medical Research Council | MR/S007555/1 | Thomas A Bowden |
| Wellcome Trust | 10.35802/203141 | Thomas A Bowden |
| National Taiwan University | NTU-CC-113L892904 | Kuan-Ying A Huang |
| National Taiwan University | NTU-CC-114L891104 | Kuan-Ying A Huang |
| Townsend-Jeantet Prize Charitable Trust | | Alain RM Townsend |

The funders had no role in study design, data collection, and interpretation, or the decision to submit the work for publication. For the purpose of Open Access, the authors have applied a CC BY public copyright license to any Author Accepted Manuscript version arising from this submission.

## Author contributions

Pramila Rijal, Conceptualization, Resources, Data curation, Formal analysis, Supervision, Funding acquisition, Validation, Investigation, Visualization, Methodology, Writing – original draft, Project administration, Writing – review and editing; Leiyan Wei, Investigation, Writing – review and editing; Guido C Paesen, Data curation, Formal analysis, Investigation, Visualization, Methodology, Writing – review and editing; David I Stuart, Supervision, Validation, Writing – review and editing; Mark Haworth, Supervision, Writing – review and editing; Kuan-Ying A Huang, Resources, Funding acquisition, Writing – review and editing, Investigation, Conceptualization, Formal analysis, Methodology, Validation; Thomas A Bowden, Formal analysis, Supervision, Funding acquisition, Validation, Visualization, Writing – review and editing; Alain RM Townsend, Conceptualization, Formal analysis, Supervision, Funding acquisition, Validation, Methodology, Writing – original draft, Writing – review and editing

## Author ORCIDs

Pramila Rijal ⓘ https://orcid.org/0000-0002-9214-9851
Guido C Paesen ⓘ https://orcid.org/0000-0003-2373-2645
Kuan-Ying A Huang ⓘ https://orcid.org/0000-0001-6891-6945
Thomas A Bowden ⓘ https://orcid.org/0000-0002-8066-8785

## Ethics

Animal experiments were conducted in strict compliance with the UK Animals (Scientific Procedures) Act Project License (PP9362617), following the principles of the 3Rs (Replacement, Reduction, and Refinement). The mice were housed in accordance with the UK Home Office ethical and welfare guidelines, fed standard chow, and had access to water ad libitum.

Reviewer #1 (Public review): https://doi.org/10.7554/eLife.105317.3.sa1
Reviewer #2 (Public review): https://doi.org/10.7554/eLife.105317.3.sa2
Author response https://doi.org/10.7554/eLife.105317.3.sa3

# Additional files

## Supplementary files

Supplementary file 1. Recombinant NA expression in various cell expression systems.

Supplementary file 2. List of loop annotations in *Varghese et al., 1983* used as a reference for creating NA hybrids in this paper. Loops in mSN1, N1/09, N1/19, and PR8N1 are listed. Text in

orange colour where N1 differs from the original N2 numbering system. Residues in the loops that differ from mSN1 sequence are marked in red ('aa' denotes amino acids).

Supplementary file 3. Crystallographic data collection and refinement statistics.

Supplementary file 4. Amino acid sequences of NA gene constructs.

MDAR checklist

## Data availability

Diffraction data have been deposited in PDB under the accession codes 9GQX, 9GQT, and 9GQQ. All data generated or analysed during this study are included in the manuscript and supporting files; source data files have been provided for Figures 3 and 5–7. *Figure 3—source data 1 and 2*, *Figure 5—source data 1 and 2*, *Figure 7—source data 1–3*, and *Figure 5—figure supplement 1—source data 1*, contain the numerical data used to generate the figures.

The following datasets were generated:

| Author(s) | Year | Dataset title | Dataset URL | Database and Identifier |
|---|---|---|---|---|
| Pramila R, Paesen GC, Bowden TA | 2024 | influenza neuraminidase mSN1 | https://doi.org/10.2210/pdb9GQX/pdb | Worldwide Protein Data Bank, 10.2210/pdb9GQX/pdb |
| Bowden TA, Paesen GC | 2024 | influenza neuraminidase hybrid N1/09 | https://doi.org/10.2210/pdb9GQT/pdb | Worldwide Protein Data Bank, 10.2210/pdb9GQT/pdb |
| Bowden TA, Paesen GC, Rijal P | 2024 | influenza neuraminidase N1/19 hybrid | https://doi.org/10.2210/pdb9GQQ/pdb | Worldwide Protein Data Bank, 10.2210/pdb9GQQ/pdb |

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

## Appendix 1

## Animal experimentation: sample size and statistical methods

### Design and endpoint

Two parallel groups (vaccinated vs control), $n = 6$ per group. The primary endpoint is time from viral challenge to humane endpoint (≥20% body-weight loss or euthanasia), with right-censoring at day 14 for animals that never meet the endpoint.

### Rationale for $n$

Historical data in this model show large group differences:

- Controls: all animals typically reach the ≥20% weight-loss endpoint within 5–7 days.
- Vaccinated: animals generally do not reach the endpoint and recover from transient weight loss by ~14 days.
- Sample size/power: with $n = 6$ per group (12 total) and an expected event fraction of ~0.5 (events concentrated in controls), the approximate powers at $a = 0.05$ are: hazard ratio (HR) = 0.30 → 31%; HR = 0.20 → 50%; HR = 0.10 → 81% (Schoenfeld approximation). Thus, the study is well-powered to detect very strong vaccine effects (HR ≤ 0.10) consistent with prior data.

With this strong historical effect, a fixed sample of 6 per group was chosen to minimise animal use while retaining power to detect such large benefits.

### Statistical analysis

- Survival curves generated in GraphPad Prism using the Kaplan–Meier method.
- Groups were compared with a two-sided logrank (Mantel–Cox) test at $a = 0.05$.

### Ethical justification (3Rs)

- Reduction: Sample size chosen as the smallest consistent with detecting the very large effects previously observed.
- Refinement: Humane endpoint (≥20% weight loss) with early euthanasia to limit suffering.
- Replacement: No suitable non-animal alternative exists for this disease model.

### Randomisation

Mice are randomly assigned to each group to ensure unbiased allocation and to minimise selection bias. The variable factor, weight of the mouse, which can affect outcome measures (such as antibody response and weight loss following virus infection), is controlled by using mice from the same age group cohort with minimal weight variation. Simple randomisation is performed to allocate mice in each group. They are randomised in advance by the Biomedical Services (BMS) staff during cage grouping, who are blinded to the experimental groups.

### Blinding

For in vivo experiments, the experimenter was not blinded during procedures or in vitro experimentation due to personnel and resources limitations. However, efforts were made to minimise bias due to unblinding.

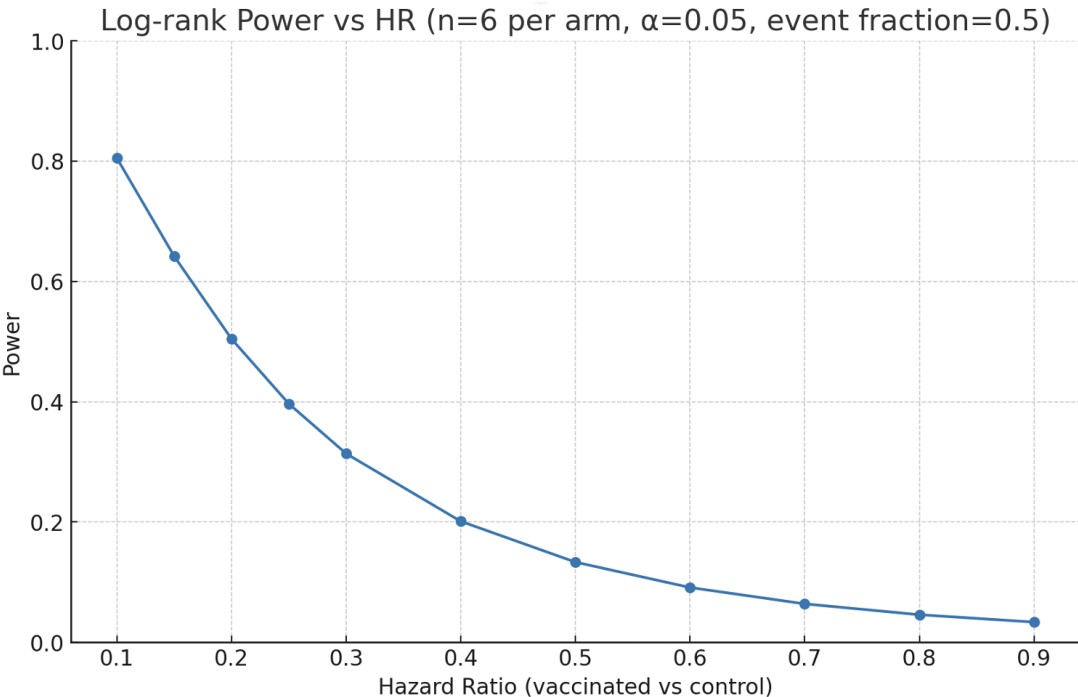

**Appendix 1—figure 1.** Power analysis for animal experiments.

