## [Editor Report · eLife Assessment]

The authors developed a methodology to graft antigenic surface loops on influenza virus neuraminidases. The hybrid proteins retained the structure of the neuraminidase scaffold and the antigenicity of the grafted loops. This **fundamental** work should help in developing novel neuraminidase constructs for use in influenza virus vaccines. The paper presents **compelling** evidence supporting the conclusions arrived at by the authors.

---

## [Referee Report · Reviewer #1 (Public review)]

Summary:

This manuscript described a structure-guided approach to graft important antigenic loops of the neuraminidase to a homotypic but heterologous NA. This approach allows the generation of well-expressed and thermostable recombinant proteins with antigenic epitopes of choice to some extent. The loop-grafted NA was designated hybrid.

Strengths:

The hybrid NA appeared to be more structurally stable than the loop-donor protein while acquiring its antigenicity. This approach is of value when developing a subunit NA vaccine which is difficult to express. So that antigenic loops could be potentially grafted to a stable NA scaffold to transfer strain-specific antigenicity.

---

## [Referee Report · Reviewer #2 (Public review)]

In their manuscript, Rijal and colleagues describe a 'loop grafting' strategy to enhance expression levels and stability of recombinant neuraminidase. The work is interesting and important.

Major points from first round of review:

(1) The authors overstress the importance of the epitopes covered by the loops they use and play down the importance of antibodies binding to the side, the edges, or the underside of the NA. A number of papers describing those mAbs are also not included.

(2) The rationale regarding the PR8 hybrid is not well described and should be described better.

(3) Figure 3B and 6C: This should be given as numbers (quantified), not as '+'.

(4) Figure 5A and 7A: Negative controls are missing.

(5) The authors claim that they generate stable tetramers. Judging from SDS-PAGE provided in Supplementary Figure 3B (BS3-crosslined), many different species are present including monomers, dimers, tetramers, and degradation products of tetramers. In line 7 for example there are at least 5 bands.

[Editors' note: the authors have appropriately responded to and addressed these points.]

---

## [Author Response]

The following is the authors’ response to the original reviews

**Public Reviews:**

**Reviewer #1 (Public review):**
Summary:This manuscript described a structure-guided approach to graft important antigenic loops of the neuraminidase to a homotypic but heterologous NA. This approach allows the generation of well-expressed and thermostable recombinant proteins with antigenic epitopes of choice to some extent. The loop-grafted NA was designated hybrid.Strengths:The hybrid NA appeared to be more structurally stable than the loop-donor protein while acquiring its antigenicity. This approach is of value when developing a subunit NA vaccine which is difficult to express. So that antigenic loops could be potentially grafted to a stable NA scaffold to transfer strain-specific antigenicity.Weaknesses:However, major revisions to better organize the text, and figure and make clarifications on a number of points, are needed. There are a few cases in which a later figure was described first, data in the figures were not sufficiently described, or where there were mismatched references to figures.More importantly, the hybrid proteins did not show any of the advantages over the loop-donor protein in the format of VLP vaccine in mouse studies, so it's not clear why such an approach is needed to begin with if the original protein is doing fine.

We thank the reviewer for their helpful comments. We have incorporated feedback from the authors to improve the manuscript. Please see our point-by-point response.

The purpose of loop-grafting between H5N1/2021 (a high-expressor) and the PR8 virus was not to improve the expression of PR8, which is already a good expressing NA. Instead, the loop-grafting and the in vivo experiments were done to show the loop-specific protection following a lethal PR8 virus challenge.

**Reviewer #2 (Public review):**
In their manuscript, Rijal and colleagues describe a 'loop grafting' strategy to enhance expression levels and stability of recombinant neuraminidase. The work is interesting and important, but there are several points that need the author's attention.Major points(1) The authors overstress the importance of the epitopes covered by the loops they use and play down the importance of antibodies binding to the side, the edges, or the underside of the NA. A number of papers describing those mAbs are also not included.

We have discussed the distribution of epitopes on NA molecule in the Discussion section "The distribution of epitopes in neuraminidase" (new line number 350). In Supplementary Figures 1 and 2, we have compiled the epitopes reported by polyclonal sera and mAbs via escape virus selection or crystal structural studies. There are 45 residues examples of escape virus selection, and we found that approximately 90% of the epitopes are located within the top loops (Loops 01 and Loops 23, which include the lateral sides and edges of NA). We have also included the epitopes of underside mAbs NDS.1 and NDS.3 in Supplementary Figure 2. Some of the interactions formed by these mAbs are also within the L01 and L23 loops. All relevant references are cited in Supplementary Figures 1 and 2.

A new figure has been added [Figure 1b (ii)] to illustrate the surface mapping of epitopes on NA.

(2) The rationale regarding the PR8 hybrid is not well described and should be described better.

We described the rationale for the PR8 hybrid (new lines 247-250). For clarity, we have added the following sentence within the section "Loop transfer between two distant N1 NAs:...."

(new lines 255-258):

"mSN1 showed sufficient cross-reactivity to N1/09 to protect mice against virus challenge. Therefore, we performed loop transfer between mSN1 and PR8N1, which differ by 18 residues within the L01 and L23 loops and show no or minimal cross-reactivity, to assess the loop-specific protection."

(3) Figure 3B and 6C: This should be given as numbers (quantified), not as '+'.

We have included the numerical data in Supplementary Figure 6. The data is presented in semi-quantitative manner for simplification. To improve clarity, we have now added the following sentence to the Figure 3c legend: "Refer to Supplementary Figure 6 for binding titration data".

(4) Figure 5A and 7A: Negative controls are missing.

A pool of Empty VLP sera was included as a negative control, showing no inhibition at 1:40 dilution. In the figure legends, we have stated "Pooled sera to unconjugated mi3 VLP was negative control and showed no inhibition at 1:40 dilution (not included in the graphs)"

(5) The authors claim that they generate stable tetramers. Judging from SDS-PAGE provided in Supplementary Figure 3B (BS3-crosslinked), many different species are present including monomers, dimers, tetramers, and degradation products of tetramers. In line 7 for example there are at least 5 bands.

Tetrameric conformation of soluble proteins is evidenced by the size-exclusion chromatographs shown in Figures 3a and 6b. The BS3 crosslinked SDS-PAGE are only suggestive data, indicating that the protein is a tetramer if a band appears at ~250 kDa. However, depending on the reaction conditions, lower molecular weight bands may also be observed if crosslinking is incomplete.

**Recommendations for the authors:**

**Reviewer #1 (Recommendations for the authors):**
Specific comments:- Description of Figure 2 on page 3 should go before Figure 3 lines 87-105 or swap the order of the two figures.

We have moved lines 91-96, which refer to Figure 3, to appear after Figure 2.

- Figure 3a, an EC50 should be calculated for both NA activity assay.

Figure 3a has been updated to include the EC50 and AUC (Area under curve) values for both NA activity assays. The same update has also been made for Figure 6b.

- Line 150, I'm not sure it's appropriate to cite a manuscript that was in preparation but not published. I'm referring to the two mAbs AG7C and AF9C that were claimed to bind to the L01 and L23 loops but not.

We have changed the "manuscript in preparation" to "personal communication with Dr. Yan Wu, Capital Medical University".

- The description in Figure 4a is lacking.

We have added a detailed description for Figure 4a.

- Figure 4c, sufficient description is needed. For example, the cavity should be outlined and annotated, what is the role of Val149? Why the first monomer is assigned a number of II and the second monomer with a number of I.

We have added a detailed description for Figure 4c and amended the figure as per the reviewer’s suggestions.

- Figure 5a, in addition to ELLA data to mSN1 and N1/09, ELLA data to N1/19 should also be measured and shown. Figure S7, please show IC50 instead of curves for better comparison.

We included IC50 for mSN1 and N1/09 as we intended to associate the loops with protection. Graphs for N1/19 have not been reported, but the IC50 titres from pooled sera are shown in Supplementary Figure 7 as a representation. Due to the limited sera sample sourced from tail vein bleed, these assays were performed using pooled sera, which represent the total response (established in numbers of experiments).

- Line 234-238, the author made a statement about the data shown in Figure 7b "These results mirrored several studies in the literature which showed that immunization with the 2009 N1 could provide at least partial protection in mice and ferrets to the avian H5N1 challenge". The data did not reflect that. In Figure 5b, mSN1 protects as well as other proteins. In fact, there was no advantage of N109 and N109 hybrid over mSN1 in protection against the homologous H1N109. Although higher levels of NAI antibodies were induced with the homologous protein in Figure 5a. The protection could be contributed by non-NAI antibodies, so the authors should measure binding antibodies. The author may increase the challenge dose from 200 LD50 to 1000 LD50 to see a difference due to the strong immunogenicity of the nanoparticles vaccine plus addavax. Otherwise, it looks like loop grafting is not necessary as heterologous NA could broadly protect.

We agree that msN1, despite its low NAI titres, was equally protective as homologous NA or its hybrid NA against H1N1/09 virus challenge at 200 LD50. There may be additional protective components, including non-NAI antibodies in homologous groups that may have contributed to the protection.

We assessed sera binding to H1N1/2009 and found that the binding antibody levels were also lower in the msN1 group. The corresponding graph has now been added in Figure S7d. It was difficult to determine the NAI titre required to confer protection in this experiment. For this reason, we later chose PR8 as the challenge virus to demonstrate loop-specific protection.

We are uncertain whether a 1000 LD50 challenge would have helped establish a correlation between protection and NAI IC50 titres, as the dose used is already lethal for DBA/2 mice.

- Why would the authors separate work with N1/09 and N1/19 from PR8 N1? To this reviewer's understanding, they are all the same strategies with increasing numbers of dissimilar residues from N1/09 (12) to N1/19 (16) and to PR8 (18). They are all characterized by the same approaches in vitro and in vivo.

We had two different goals for making hybrids with N1/09 and PR8 N1, therefore, we have presented these results separately.

(1) For N1/09 and N1/19, we showed that loop-grafting improved protein yield and stability. Additionally, we showed that the N1/09 hybrid can be as protective as the homologous protein.

(2) PR8 N1 is a high-yielding protein, so loop grafting did not significantly increase its yield. However, the PR8 virus challenge confirmed loop-specific protection.

- For in vivo study testing the PR8 construct, although PR8 and PR8 hybrid protect better than the heterologous mSN1, the hybrid again did not show any advantages over the PR8 original proteins.

That's correct - the PR8 hybrid was not advantageous over the original PR8 protein. However, the purpose of this experiment was to demonstrate loop specific protection. The PR8 hybrid (PR8 loops - mS scaffold) protected 6/6 mice, whereas mS hybrid (mS loops - PR8 scaffold) provided no protection.

- Line 243-249, lack of reference to figures.

References to Supplementary Figure 7b,c and Figure 2 has been added.

- What was the reason that the challenge was one by 200 LD50 for 2009 H1N1 and 1000 LD50 for PR8.

Viruses were titrated in the BALB/c strain for PR8 virus and the DBA/2 strain for X-179A (H1N1/2009) virus. These doses were selected based on their lethality and the time required to reach the endpoint (~20% weight loss) post-infection, which is 5-6 days. Most studies in the literature have used 10 LD50 or higher; thus the virus doses we used are relatively high.

- Line 268, there is no Figure 5C.

This was a mistake and has been corrected to Figure 6c.

- Line 275 what are the readers supposed to see in supplementary Figure 5a? There is not enough description for the referred figures.

A sentence has been added to Fig S5a description, to make a point about recognition of the NA scaffold by mAb CD6. "Binding by mAb CD6 is predominantly scaffold dependent and occurs across two protomers"

- The discussion is very long and some of it is not relevant to the study. For example, the role of the tetramerization domain and the basis for structurally stable tetramer formation, were not the focuses of this study.

We felt it was important to discuss the tetramerisation domain and the basis for stable tetramer formation. A previous study by Ellis *et al.* used the VASP tetramerisation domain and introduced multiple NA interface mutations to achieve a more stable closed conformation. In contrast, NA proteins used in our study required the tetrabrachion tetramerisation domain to form a properly assembled tetramer.

In lines 382-383, there is one unfinished sentence.

This is corrected.

The definition of the loops is also confusing. Line 381, the author stated that in the N1/19 hybrid design, residue N200S, could have been considered as part of the loop B2L23, and was it not?

The designation of loop ends should not be rigid but rather based on multiple factors such as, their proximity to antigenic epitopes, charge, and hydrophobicity. This is discussed in the " Definition of loops" section.

- Figure 1a and Figure S2, please provide sufficient descriptions, what do the blocks in different colors mean?

We have updated the Figure 1a legend to indicate the colours.

The descriptions for Figures S1 and S2 have also been revised for clarity.

**Reviewer #2 (Recommendations for the authors):**
Minor points(1) Line 37: Should be 'Influenza virus neuraminidase'.

This is corrected.

(2) Line 65: https://pubmed.ncbi.nlm.nih.gov/35446141/, https://pubmed.ncbi.nlm.nih.gov/33568453/ and https://pubmed.ncbi.nlm.nih.gov/28827718/ indicate that protective mAbs bind all over the NA head domain.

We have discussed the epitopes on the NA head in detail in the section "The distribution of epitopes on Neuraminidase". In Supplementary Figures 1 and 2, we compiled several studies, including those on polyclonal sera and mAbs epitopes, emphasizing that loops 01 and 23 are the predominant antibody targets (~90%). Some antibodies also bind to the underside of NA. We have discussed and referenced these studies accordingly.

A new figure has been added [Figure 1b (ii)] to illustrate the surface mapping of epitopes on NA.

The first reference has been included in both our discussion and Supplementary figure 1.

The NA epitopes discussed in the second reference have also been incorporated into our discussion and Supplementary figures 1 and 2. Note that, the E258K mutation generated on the NA underside was not relevant to mAbs and was generated randomly by passaging of H3N2 A/New York/PV190/2017 virus.

The third reference pertains to murine mAbs against influenza B virus NA.

(3) Lines 71, 72, and throughout: 'et al.' should be in italics.

All "et al." have been italicised.

(4) Many abbreviations are not defined including CHO, SDS-PAGE, MUNANA, mi3, HEPES, BSA, TPCK, MWCO, HRP, PBS, TMB, TCID50, LD50, MES, PEG, PGA, MME, PGA-LM.

The text has been amended to define these abbreviations.

(5) Line 209: Shouldn't this be ID50 instead of IC50? Also, it is not defined.

IC50 has been defined.

(6) Line 210, line 346, line 581-582: No need to capitalize letters at the beginning of words mid-sentence.

This is amended.

(7) Line 227: Is 2009 H1N1 NA meant?

This has been changed to "H1N1/2009 neuraminidase"

(8) Line 310: Is this really quantitatively true? (see major comment 1).

Based on the compilation of epitopes from published NA mAbs and polyclonal sera (via escape mutagenesis and NA-Fabs crystal structures), it is accurate to state that the protective epitopes are primarily located within loops 01 and 23.

Please also refer to our response to minor point 2.

(9) Line 352 and throughout the manuscript: 'in vitro' should be in italics.

This is amended.

(10) Line 355: https://pubmed.ncbi.nlm.nih.gov/35446141/, https://pubmed.ncbi.nlm.nih.gov/33568453/ and https://pubmed.ncbi.nlm.nih.gov/28827718/ should be included here.

Studies reporting epitopes on Influenza A neuraminidase have been compiled in Supplementary Figures 1 and 2 and cited appropriately.

(11) Line 365: https://pubmed.ncbi.nlm.nih.gov/35446141/ and https://pubmed.ncbi.nlm.nih.gov/33568453/ also describe epitopes on the underside of the NA.

Please refer to the above response to point 10.

(12) Line 365: Reference https://pubmed.ncbi.nlm.nih.gov/37506693/ is missing here.

The reference has been added.

(13) Line 369-371: Is it really a minority?

In terms of the *protective* response, the majority of the antibody response is directed towards loops 01 and 23, which form the top antigenic surface. The term 'lateral' is used in some literature to describe NA mAb epitopes; loops 01 and 23 also encompass the lateral regions.

To clarify this, we have added the following sentence to the Discussion section - "The distribution of epitopes on neuraminidase"

"It is important to note that loops 01 and 23 include a portion of epitopes that have been described in the literature as side, lateral, or underside (see mAbs NDS.1, NDS.3, and CD6 in Supplementary Fig. 2)"

Additionally in our studies in mice, we showed that protection is mediated by antibodies targeting the loops (Figure 7). We are uncertain about the binding response to the NA underside, but the NA inhibiting and protective response to the underside appears to be minimal.

Furthermore Lederhof et al. showed that among the 'underside' mAbs, NDS.1 protected mice against virus challenge, whereas NDS.3 did not. In our analysis (Supplementary Figure 2), NDS.1 makes eight-residue contacts with B4L01 and B5L01, whereas NDS.3 make five-residue contacts with B3L01 and B4L01.

(14) Line 530: The A in ELLA already stands for assay.

This is corrected.